# Closing the loop in medical decision support by understanding clinical decision-making: A case study on organ transplantation

**Yuchao Qin**[*]
University of Cambridge
yq257@cam.ac.uk

**Fergus Imrie**[*]
University of California, Los Angeles
imrie@g.ucla.edu

**Alihan Hüyük**
University of Cambridge
ah2075@cam.ac.uk

**Daniel Jarrett**
University of Cambridge
daniel.jarrett@maths.cam.ac.uk

**Alexander Edward Gimson**
University of Cambridge
alexander.gimson@nhs.net

**Mihaela van der Schaar**
University of Cambridge
The Alan Turing Institute
University of California, Los Angeles
mv472@cam.ac.uk

## Abstract

Significant effort has been placed on developing decision support tools to improve patient care. However, drivers of real-world clinical decisions in complex medical scenarios are not yet well-understood, resulting in substantial gaps between these tools and practical applications. In light of this, we highlight that more attention on understanding clinical decision-making is required both to elucidate current clinical practices and to enable effective human-machine interactions. This is imperative in high-stakes scenarios with scarce available resources. Using organ transplantation as a case study, we formalize the desiderata of methods for understanding clinical decision-making. We show that most existing machine learning methods are insufficient to meet these requirements and propose iTransplant, a novel data-driven framework to learn the factors affecting decisions on organ offers in an instance-wise fashion directly from clinical data, as a possible solution. Through experiments on real-world liver transplantation data from OPTN, we demonstrate the use of iTransplant to: (1) discover which criteria are most important to clinicians for organ offer acceptance; (2) identify patient-specific organ preferences of clinicians allowing automatic patient stratification; and (3) explore variations in transplantation practices between different transplant centers. Finally, we emphasize that the insights gained by iTransplant can be used to inform the development of future decision support tools.

## 1 Introduction

The decision that a patient and a clinician jointly make when a donor organ is offered for transplantation is critical and carries serious consequences. Even though the initial offer of a donor organ for a particular recipient is made on the basis of agreed criteria as to optimum allocation

---

[*]Equal contribution

35th Conference on Neural Information Processing Systems (NeurIPS 2021).

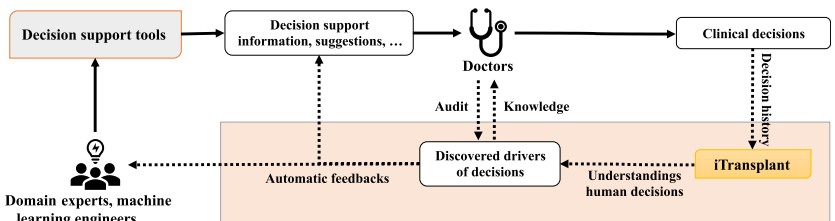

Figure 1: Closing the loop of medical decision support by understanding clinical decision-making. Most existing clinical decision support tools offer generic advice or alerts to clinicians without specificity for the immediate decision about to be taken. In this paper, we highlight that iTransplant, by identifying drivers of that decision, will be able to feedback into future iterations of the decision support tool specific information about which factors impacted the decisions so that in future that specific information can be given to the decision maker or can be taken into account in designing future iterations of the decision support tool itself.

within each transplant jurisdiction, there remains substantial variation in the rate at which organs are accepted. Clinical variation is a well-observed phenomenon, but may have profound consequences and unfavourably impact outcomes. Addressing clinical variation is therefore a major priority in many healthcare systems. Understanding the factors which are associated with variation in clinical decision-making is an important goal as it might be able to inform clinicians of biases which, if rectified, might result in improved clinical outcomes. With a case study on organ transplantation, we explore the potential of inverse decision-making approaches to shed light on such factors.

**Organ transplantation**    Transplantation is typically the last life-saving treatment available for patients with end-stage diseases that cause organ failures. However, due to the limited availability of donors, patients often have to wait years before transplantation [14, 23]. Regrettably, waitlists continue to grow despite increases in the number of donors and many patients die while waiting for an organ, with over 7,500 deaths each year in the United States alone [20]. The majority of these patients received at least one organ offer that was declined on their behalf [12], despite these organs often appearing to be suitable for transplantation and yielding good outcomes when eventually transplanted [17]. It is therefore important to understand why donor organs subsequently successfully implanted have been declined for previous patients.

**Clinicians' decision-making is poorly understood**    When a donor organ becomes available, it is first offered to a patient on the waitlist on the basis of agreed offering criteria. Once an organ is offered, a clinician must choose whether to accept or decline the organ offer. Although significant effort has been placed into developing organ allocation algorithms [47, 28, 4], a donor offered to the first ranked patient in a waitlist is rejected up to 50% of the time [12, 43].

In real-world organ transplantation systems, the performance of organ allocation algorithms are significantly affected by clinicians' assessments of organ offers. Even for good quality organs, the assigned organ offers could be turned down several times before they are finally accepted [45, 17]. The high ratio of declined organ offers is important as it may impact outcomes for that organ (e.g. due to prolonged cold ischemia time) and the patients involved (e.g. [12] shows that centers with higher acceptance rates experienced significantly lower adjusted estimated waitlist mortality of the highest-ranked patients).

**Substantial variation in clinical practice**    Variation in clinical practice is an extensively studied phenomenon across medicine with significant impacts on organ transplantation [42, 2]. Striking discrepancies in organ offer acceptance rates have been observed for different transplant centers, even after accounting for organ quality and the severity of the recipient's illness [12]. However, it has been impossible to disambiguate the causes of this variation, despite its importance for understanding current medical practices and ultimately improving organ allocation policy.

In this paper, we highlight that to address such challenges it is necessary for practical inverse decision-making approaches to provide interpretable and personalized insights into clinical decision-making.

**Human interpretable policies**    Interpretability and transparency are crucial for machine learning applications in medicine [39]. As a result, white-box models, such as logistic regression, are widely adopted in the medical literature (e.g. [33, 10]). Numerous studies have demonstrated

improved performance from using black-box models for medical applications [15], albeit at the cost of interpretability. The development of interpretable, yet highly performant, models for clinical decision-making is essential to bridge the gap to black-box models and further medical knowledge. In addition, for such interpretations to be useful for understanding clinical decision-making, models must be counterfactually consistent (i.e. the counterfactual prediction must match the interpretation).

**Precision medicine** Precision, or personalized, medicine seeks to improve medical care by tailoring therapy to the needs of a particular patient [40]. However, in the organ transplantation setting, most existing methods only learn a global decision-making policy for all patients, which fails to address the discrepancies in clinicians' policies for patients from different cohorts (see, e.g., [16, 5, 35]). To both understand clinicians' decision-making and improve precision medicine, models that learn personalized policies are necessary.

**Our contributions** In this paper, we propose *iTransplant (individualized TRANSparent Policy Learning for orgAN Transplantation)*, a novel data-driven framework to learn interpretable organ offer acceptance policies directly from clinical data. Our method learns a patient-wise parametrization of the expert clinician policy that accounts for the differences between patients, a crucial but often overlooked factor in organ transplantation.

We achieve this by training a neural network-based policy selector to identify individualized policies for patients from different cohorts. These policies act on the space of known match criteria using a white-box function, ensuring interpretability with respect to the match criteria. Our method significantly outperforms existing interpretable models, with comparable accuracy to black-box approaches.

We conduct several investigative experiments with real-world liver transplantation data from the Organ Procurement and Transplantation Network (OPTN), covering 190,525 organ offers. The results show that iTransplant can be used to probe clinical decision-making practices in a number of ways. Our investigations allow us to: (1) identify important match criteria for organ offer acceptance; (2) discover patient-wise organ preferences of clinicians via automatic patient stratification in a latent representation space; and (3) examine the transplantation practice variations across transplant centers.

## 2 Problem Formulation

**Notation** We denote the feature space of all possible patients as $\mathcal{X} \subseteq \mathbb{R}^d$ and the feature space of all possible organs available as $\mathcal{O} \subseteq \mathbb{R}^e$. The organ offer that assigns organ $\mathbf{O} \in \mathcal{O}$ to patient $\mathbf{X} \in \mathcal{X}$ is denoted with $\mathbf{s} = (\mathbf{X}, \mathbf{O})$ and the associated decision of clinicians is denoted as $a \in \{0, 1\}$. Here, the event of offer acceptance is denoted as $\{a = 1\}$, and $\{a = 0\}$ means offer rejection. The decision-making policy of clinicians (expert policy) is assumed to be a probability distribution $\pi^*(a|\mathbf{s})$ conditioned on the organ offer $\mathbf{s}$.

Following the discussion on interpretability and precision medicine in Section 1, we propose three key desiderata of practical inverse decision-making approaches: 1) personalized policies, 2) interpretable insights of decisions, and 3) consistent interpretations under perturbation. First, let us introduce some concepts and assumptions related to the above requirements.

**Criteria space** Suppose there are $l$ known match criteria related to decisions on organ offers, denoted by a set of functions $\mathcal{C} = \{c_i(\mathbf{s}) \colon \mathcal{X} \times \mathcal{O} \to \mathbb{R}, i = 1, 2, \ldots, l\}$. Each criterion $c_i \in \mathcal{C}$ takes patient and organ features as input and generates a match score as the assessment of organ offer $\mathbf{s} = (\mathbf{X}, \mathbf{O})$. Based on $l$ match criteria in $\mathcal{C}$, we define a transform $\mathcal{T} \colon \mathcal{X} \times \mathcal{O} \to \mathcal{M} \subseteq \mathbb{R}^l$ that maps organ offers to a criteria space $\mathcal{M}$. Given an organ offer $\mathbf{s}$, its representation in space $\mathcal{M}$ can be calculated as $\mathcal{T}(\mathbf{s}) \coloneqq [m_1(\mathbf{s}), m_2(\mathbf{s}), \ldots, m_l(\mathbf{s})]'$. In practice, the criteria space would usually be proposed by experts based on domain knowledge. Note, our method allows clinicians to explore criteria space $\mathcal{M}$ with different sets of criteria based on their own expertise.

In line with our desideratum for a personalized policy, we have Assumption 1 that bridges the criteria space $\mathcal{M}$ with the decision-making policy of clinicians on a per patient basis.

**Assumption 1** (Partial monotonicity) There exists a vector $\mathbf{v} \in \{-1, 1\}^l$ such that for any two organ offers $\mathbf{s}_1 = (\mathbf{X}_1, \mathbf{O}_1), \mathbf{s}_2 = (\mathbf{X}_2, \mathbf{O}_2)$ satisfying the conditions of $\mathbf{X}_1 = \mathbf{X}_2$ and $\mathbf{v} \circ \mathcal{T}(\mathbf{s}_1) \succeq \mathbf{v} \circ \mathcal{T}(\mathbf{s}_2)$, where $\circ$ is the Hadamard product operator, we have $\pi^*(a = 1|\mathbf{s}_1) \geq \pi^*(a = 1|\mathbf{s}_2)$, where $\pi^*$ is clinicians' decision-making policy on organ offers.

**Assumption 2** (Greedy decision policy) Noting that decisions on organ offers by clinicians will not necessarily affect the next organ offer assigned to their patients, we further assume that decisions of clinicians are consistent with a greedy policy, i.e., maximizing the immediate benefit $R(\mathbf{s}, a)$ of decision $a$ on organ offer $\mathbf{s}$ with policy $\pi^*(a|\mathbf{s}) = \arg\max_{a \in \{0,1\}} R(\mathbf{s}, a)$.

Related to the requirement for a personalized policy, clinicians may use different subsets of criteria in $\mathcal{M}$ to evaluate potential outcomes of the offered organ for different cohorts of patients. For instance, [5] reports that the difference in age of donor and recipient has diverse impacts on post-transplant mortality, and young recipients with elderly donors are most affected. Hence, a personalized reward structure is necessary to account for policy variations at the patient level, which leads to the following assumption:

**Assumption 3** (Personalized rewards) Similar to existing literature on inverse decision-making (e.g., [49, 30]), we assume a linear structure of the reward function: $R(\mathbf{s}, a) = \langle \rho_a^*(\mathbf{X}), \mathcal{T}(\mathbf{s}) \rangle$, where $\langle \cdot, \cdot \rangle$ is the inner product of two vectors in an Euclidean space, $\rho_a^*(\mathbf{X})$ is the weight vector for different criteria in space $\mathcal{M}$. Note that there exists a family of equivalence reward functions $\mathcal{R}(\mathbf{s}, a) = R(\mathbf{s}, a) - \Phi(\mathbf{s})$, where $\Phi(\mathbf{s}) : \mathcal{X} \times \mathcal{O} \mapsto \mathbb{R}$ can be arbitrary scalar functions of $\mathbf{s}$ (see, e.g., [29]), that are consistent with expert policy $\pi^*$. For the sake of convenience, we assume $R(\mathbf{s}, a = 0) \equiv 0$ and $R(\mathbf{s}, a = 1) = \langle \rho^*(\mathbf{X}), \mathcal{T}(\mathbf{s}) \rangle$. It is worth noting that $\rho^*(\mathbf{X})$ is a function of patient features $\mathbf{X}$ and thus is able to represent the patient-specific rewards for clinicians.

To ensure that the decision policy is differentiable, we further adopt the maximum entropy assumption [49] that $\pi^*$ follows a Boltzmann distribution: $\pi^*(a|\mathbf{s}) \propto \exp[R(\mathbf{s}, a)]$, which is a *soft* version of the $\arg\max$ operator. Note that expert policy $\pi^*$ is uniquely determined by the true reward parameter map $\rho^*(\mathbf{X})$, in this paper, we seek to learn a representation $\hat{\pi}$ of expert policy $\pi^*$ with all three desiderata achieved by leveraging the notion of a transparent policy space introduced as follows.

**Transparent policy space**  Following the Boltzmann distribution formulation of expert policy $\pi^*$, we can construct a transparent policy space $\Pi$ as

$$\Pi = \{\text{Bernoulli}(p) \colon p = \frac{1}{1 + \exp\left[-\langle \rho, \mathcal{T}(\mathbf{s}) \rangle\right]}, \rho \in \mathbb{R}^l\}, \tag{1}$$

where $\mathcal{T}(\mathbf{s}) \in \mathcal{M}$. We call the vector $\rho \in \mathbb{R}^l$ a *policy signature* since it uniquely characterizes the behavior of the corresponding policy $\pi_\rho$ in space $\Pi$. Due to the interpretability of criteria space $\mathcal{M}$ and the linear structure in equation (1), policies in space $\Pi$ are human comprehensible by design and are considered *transparent*. Note that for the same patient feature vector $\mathbf{X} \in \mathcal{X}$, there exists a policy signature $\rho_{\mathbf{X}} = \rho^*(\mathbf{X})$ such that $\pi^*(a|\mathbf{X}, \mathbf{O}) = \pi_{\rho_{\mathbf{X}}} \in \Pi, \forall \mathbf{O} \in \mathcal{O}$. Thereby, the expert policy $\pi^*$ can be represented with patient-wise projections in space $\Pi$.

**Personalized policy projection**  Given demonstrations from expert policy $\pi^*$ and criteria space $\mathcal{M}$, our target is to find a patient-wise projection $\hat{\pi}^{(\theta)}$ of policy $\pi^*$ in the transparent policy space $\Pi$ such that the distance $\mathrm{d}(\pi^*, \hat{\pi}^{(\theta)})$ is minimized via $\min_\theta \mathrm{d}(\pi^*, \hat{\pi}^{(\theta)})$, where $\hat{\pi}^{(\theta)} = \pi_{\rho(\mathbf{X};\theta)}$ and $\rho(\mathbf{X}; \theta)$ is a function of patient feature vector $\mathbf{X}$ with learnable parameter set $\theta$. The distance $\mathrm{d}(\pi^*, \hat{\pi}^{(\theta)})$ is defined as the accumulated Kullback–Leibler (KL) divergence $\mathrm{d}(\pi^*, \hat{\pi}^{(\theta)}) \coloneqq \mathbb{E}_{\mathbf{s} \sim \triangle(\mathcal{X} \times \mathcal{O})}[D_{\mathrm{KL}}(\pi^* \| \hat{\pi}^{(\theta)})|\mathbf{s}]$, where $\triangle(\mathcal{X} \times \mathcal{O})$ is the organ offer distribution. The requirement for consistency of $\hat{\pi}^{(\theta)}$ is addressed in Section 3.

## 3 Individualized Transparent Policy Learning

In this paper, we propose a data-driven framework, iTransplant, for learning patient-wise policies that match clinical practice. We utilize neural networks as general function approximators for the policy signature $\rho(\mathbf{X}; \theta)$. The architecture of the proposed model is illustrated in Figure 2.

While the idea in [22] of mapping the input features into a concept space and performing prediction tasks using the concept space is in philosophy similar to our proposed method, the target of our method differs significantly from [22]. [22] aims to predict a set of human-specified concepts as an intermediate step, with each concept having a fixed (albeit learned) importance on the final prediction. Contrastingly, iTransplant aims to predict individualized policies (equivalently a set of weights) over a set of clinician-specified criteria, but not the fixed values of such criteria. The distinct targets lead to significant differences in the problem formulation, methodology and analysis in our paper and in [22].

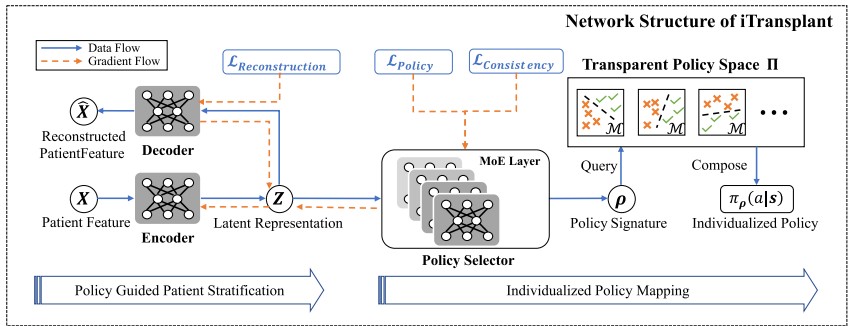

Figure 2: Network structure of the iTransplant framework.

In the proposed iTransplant framework, organ offers $\mathbf{s} = (\mathbf{X}, \mathbf{O})$ are mapped to criteria space $\mathcal{M}$ via a transform $\mathcal{T}$ specified by domain experts. The transparent policies identified by iTransplant are all generated based the match criteria in space $\mathcal{M}$. To achieve individualized policy identification, an auto-encoder structure is utilized to learn the latent representation $\mathbf{Z} \in \mathcal{Z}$ of patient features $\mathbf{X}$. From $\mathbf{Z}$, a specific policy signature $\rho$ is generated by the policy selector network. Finally, the individualized policy $\pi_\rho(a|\mathbf{s})$ is retrieved from the transparent policy space $\Pi$ via the policy signature $\rho$. This design ensures that the policies learned by iTransplant are innately individualized and human comprehensible. Detailed descriptions of the network structure and loss functions can be found in the Appendix.

**Policy guided patient stratification**   As shown in Figure 2, the policy selector network is built on top of a Mixture-of-Experts (MoE) layer [36], which contains a gating network and $K$ expert networks. Based on the latent representation $\mathbf{Z}$ of a patient, the gating network will select $k$ experts among the total $K$ candidates and combine their outputs as the policy signature $\rho$. The MoE structure is applied to encourage the encoder network to group patients sharing the same decision policy in the latent space (see the Appendix for illustrations). The MoE layer in the policy selector network passes the gradient from the policy projection loss $\mathcal{L}_{Policy}$ back to the encoder network, enabling it to guide the encoder to map patients share the same policy as neighbours in the latent space $\mathcal{Z}$. In this sense, in iTransplant, the representation learning of patient features in the latent space is guided by the learned policy, which allows the encoder network to learn the implicit stratification of patients from real-world transplantation data.

**Individualized policy mapping**   Our main target is to find the patient-wise policy projection $\hat{\pi}^{(\theta)} = \pi_{\rho(\mathbf{X};\theta)} \in \Pi$ of expert policy $\pi^*$ such that the distance $\mathrm{d}(\pi^*, \hat{\pi}^{(\theta)})$ is minimized. Note that the entropy of expert policy $\pi^*$ is a constant, we have $\mathrm{d}(\pi^*, \hat{\pi}^{(\theta)}) = -\mathbb{E}_{\mathbf{s}\sim\triangle(\mathcal{X}\times\mathcal{O})}[\sum_{a\in\{0,1\}} \pi^*(a|\mathbf{s})\log(\hat{\pi}^{(\theta)}(a|\mathbf{s}))] + \mathrm{Constant}$, where the first term is the accumulated cross-entropy between $\pi^*$ and $\hat{\pi}^{(\theta)}$. Given demonstrations $\mathcal{D} = \{(\mathbf{s}_i, a_i) : i = 1, 2, \ldots, N\}$ of expert policy $\pi^*$, the minimization of such accumulated KL-divergence can be achieved by minimizing the following policy projection loss $\mathcal{L}_{Policy} = -\frac{1}{N}\sum_{i=1}^{N}\sum_{a\in\{0,1\}} \mathbb{I}(a_i = a)\log(\hat{\pi}^{(\theta)}(a_i|\mathbf{s}_i))$.

**Enforcing the consistency requirement**   Note that policies in space $\Pi$ are innately consistent under perturbations to organ features. To meet the consistency requirement for inverse decision-making methods proposed in Section 2, a regularization term defined in equation (2) is enforced to improve consistency of policy projection in $\Pi$ for patients with similar latent space representations.

$$\mathcal{L}_{Consistency} = \frac{\lambda}{2}\mathbb{E}_{\mathbf{x}_1,\mathbf{x}_2\sim\triangle(\mathcal{X}\times\mathcal{X})}\left[\frac{\|\rho_1 - \rho_2\|_2^2}{\|\mathbf{z}_1 - \mathbf{z}_2\|_2^2}\bigg|\mathbf{x}_1, \mathbf{x}_2\right], \tag{2}$$

where $\triangle(\mathcal{X}\times\mathcal{X})$ is the patient pair distribution over space $\mathcal{X}\times\mathcal{X}$, $\rho_1, \rho_2$ and $\mathbf{z}_1, \mathbf{z}_2$ are policy signatures and latent representations for patients $\mathbf{x}_1$ and $\mathbf{x}_2$, respectively, $\lambda > 0$ is a hyperparameter.

# 4   Related Work

**Medical literature on organ offer acceptance**   A number of works have attempted to discover variables related to organ offer acceptance or rejection [27, 46] and to improve these decisions [41].

For instance, [33] used logistic regression models to identify donor features related to a higher risk of liver allograft discard and proposed a discard risk index that could be used to improve existing organ allocation policies. This donor discard index describes donor factors but does not address the reasons why a specific center may or may not reject a donor organ.

Table 1: Comparison with Related Work. iTransplant satisfies three key desiderata for understanding clinicians' decision-making for organ transplant offers: (1) individualized policies; (2) interpretability; (3) consistency of interpretation. Organ offers, decisions of clinicians, patient feature vector, criteria space for policy generation, expert policy, recovered policy, feature selection function and learned contributions of different criteria to decisions are denoted as $\mathbf{s}$, $a$, $\mathbf{X}$, $\mathcal{M}$, $\pi^*$, $\hat{\pi}$, $\mathcal{S}$ and $\hat{\rho}$, respectively.

| APPROACH | EXAMPLE REF | INPUT | OUTPUT | (1) | (2) | (3) |
|---|---|---|---|---|---|---|
| IMITATION LEARNING | [38] | $\{\mathbf{s}_t, a_t\} \sim \pi^*$ | $\hat{\pi}$ | ✗ | ✗ | ✗ |
| GLOBAL VARIABLE SELECTION | [31] | $\{\mathbf{s}_t, a_t\} \sim \pi^*$ | $\hat{\pi}, \mathcal{S}$ | ✗ | ✗ | ✗ |
| INSTANCE-WISE VARIABLE SELECTION | [48] | $\{\mathbf{s}_t, a_t\} \sim \pi^*$ | $\hat{\pi}, \mathcal{S}(\mathbf{s})$ | ✓ | ✗ | ✗ |
| INVERSE REINFORCEMENT LEARNING[1] | [49] | $\{\mathbf{s}_t, a_t\} \sim \pi^*, \mathcal{M}$ | $\hat{\pi}, \hat{\rho}$ | ✗ | ✓ | ✗ |
| ITRANSPLANT | (OURS) | $\{\mathbf{s}_t, a_t\} \sim \pi^*, \mathcal{M}$ | $\hat{\pi}, \hat{\rho}(\mathbf{X})$ | ✓ | ✓ | ✓ |

In this paper, we also seek to identify key variables and criteria that inform clinical decisions on organ offers. Therefore, our method is primarily designed as *a tool for understanding* the different factors or the weight of those factors between decision-making of clinicians rather than a model to describe commonly rejected donors. Our method learns clinicians' policies at the patient level rather than a global policy, enabling our method to identify different policies for different patient cohorts.

Because the information derived from iTransplant is specific to a donor offered to a center, it may act as a better prompt to de-bias clinical decision-making. Prior attempts to de-bias clinical decision-making have involved generic prompts relating to common errors in decision-making [1, 7, 8] and have been met with variable success. In contrast, our approach identifies specific factors that are involved in individual donor organ – patient acceptance decisions.

**Interpretation by feature-selection** Both global [13, 31] and instance-wise [48] feature selection methods could be applied to identify important variables involved in organ offer decisions. In comparison, iTransplant is, by design, capable of discovering important variables for decisions on organ offers at both the global and patient level. In addition, unlike methods which represent policies directly by neural networks, our method offers interpretability via the transparent policy space $\Pi$, which is critical for high-stake scenarios such as decision-making for organ offers.

**Imitation learning with post-hoc interpretation** Imitation learning (IL) [18] seeks to replicate expert polices from observational data of decision-making. In the one-step decision-making scenario considered in this paper, behavioral cloning (BC) [3, 38] could be directly applied to imitate the expert policy. However, as a black-box model, the neural networks utilized in BC provide no directly human comprehensible interpretations to the learned policy and have to rely on post-hoc interpretability methods like LIME [34] and SHAP [26] to provide insight for their predictions.

**Inverse reinforcement learning** Compared to imitation learning, inverse reinforcement learning (IRL) techniques [32, 30] seek to infer a reward function that is consistent with observed expert behaviors. The learned reward function can then be used for explanation of expert behaviors. For instance, with feature maps from domain knowledge and the assumption of linear reward structure, maximum entropy IRL [49] can effectively reveal contributions of different features to expert behaviors. Note that in the one-step decision-making setting considered in this paper, maximum entropy IRL [49] degenerates to a global logistic regression model, which means that it is unable to capture the variations in decisions made for different subgroups of patients.

---

[1]In the setting considered in this paper, IRL methods with assumptions of maximum-entropy policy and linear reward structures degenerate to logistic regression. In this case, their interpretations become consistent under perturbations. However, many IRL methods would be inconsistent in more general settings.

A comparison of iTransplant with alternate methods that could be applied to discover potential drivers of clinicians' decisions in the organ transplantation setting is provided in Table 1.

## 5 Illustrative Examples

To demonstrate the advantage of the proposed individualized policy learning framework and showcase potential applications of iTransplant as a tool for discovering variations in transplantation practices, real-world liver transplantation data from OPTN are utilized to provide several illustrative examples. In all examples included in this section, the criteria space $\mathcal{M}$ is constructed from a set of match criteria manually specified based on domain knowledge, and a detailed explanation of these match criteria can be found in the Appendix, together with full experimental details.

### 5.1 Model Validation

We first sought to validate the ability of iTransplant to predict organ offer acceptance using organ offer data from ten transplant centers with sufficient number (over 1,000) of accepted offers after removal of missing data. We compared iTransplant to a variety of baselines encompassing the approaches outlined in Table 1, including logistic regression, a white-box model frequently used in the medical literature, and neural network-based behavioral cloning (BC). Despite being a black-box model, and thus unsuitable to achieve our goal of *understanding* decision-making, we include BC as an upper bound on predictive performance using the proposed match criteria. We measure performance using three metrics: area under the receiver operating characteristic curve (AUC-ROC), area under the precision recall curve (AUC-PRC) and log-likelihood (LL) of observed samples. Hyperparameters for all methods can be found in the Appendix.

Table 2: Benchmark of different methods on decision prediction.

| METHOD | AUC-ROC | AUC-PRC | LL |
|---|---|---|---|
| LOGISTIC REGRESSION | 0.794±0.054 | 0.341±0.061 | -0.538±0.051 |
| PER-CLUSTER LOGISTIC REGRESSION | 0.803±0.049 | 0.352±0.063 | -0.527±0.048 |
| DECISION TREE | 0.775±0.057 | 0.274±0.069 | -0.552±0.060 |
| PER-CLUSTER DECISION TREE | 0.773±0.052 | 0.281±0.066 | -0.564±0.064 |
| LOCALLY WEIGHTED REGRESSION | 0.865±0.044 | 0.429±0.066 | -0.256±0.089 |
| LASSO | 0.777±0.064 | 0.314±0.068 | -0.570±0.045 |
| RANDOM FOREST | 0.852±0.064 | 0.421±0.106 | -0.271±0.092 |
| INVASE | 0.790±0.062 | 0.341±0.071 | -0.541±0.064 |
| BEHAVIORAL CLONING | 0.899±0.043 | 0.502±0.067 | -0.383±0.067 |
| ITRANSPLANT (OURS) | 0.895±0.045 | 0.502±0.062 | -0.396±0.069 |

As shown in Table 2, iTransplant significantly outperforms all baseline methods and performs similarly to the upper bound provided by BC while maintaining interpretability of policies identified for each patient. Although the policy learned by iTransplant uses logistic regression as the decision function, by learning individualized policies, iTransplant greatly outperformed both global logistic regression and per-cluster logistic regression (see Appendix for further details).

While BC and INVASE are unable to provide direct/sufficient insight into clinical decision-making (our primary goal), post-hoc interpretations could offer an alternate approach. To assess the suitability of such interpretations, we evaluated the consistency of the interpretations of iTransplant and BC/INVASE with post-hoc interpretation via LIME [34]. If a feature has a positive weight, increasing the value of the feature should increase the probability of acceptance. Indeed, this is a requirement for the interpretation to be useful. Thus we performed perturbations to single patient or organ features (25% of a standard deviation for continuous variables and flipped value for binary ones) and measured the consistency of sign between the actual deviation in prediction and the expected deviation from counterfactual interpretation (via importance of perturbed feature) on 200 organ offers.

Table 3: Consistency of interpretations.

| METHOD | CONSISTENCY |
|---|---|
| BC WITH LIME | 46.8% |
| INVASE WITH LIME | 41.7% |
| ITRANSPLANT | 85.0% |

Table 3 shows that iTransplant is highly consistent while BC/INVASE with post-hoc interpretation via LIME [34] barely exceeds random. As a result, iTransplant can be used to understand clinical decision-making, while the low consistency of post-hoc interpretations for BC and INVASE render them inappropriate for this purpose.

iTransplant benefits from the learning capability of neural networks to perform comparably with black-box models while maintaining interpretability of the learned decision-making policies. This makes iTransplant a powerful tool to investigate the decision-making policies in organ transplantation.

## 5.2 Investigative Experiments

As shown above, iTransplant can effectively learn transparent policies from the decision-making history of clinicians while achieving significant performance improvements to white-box models. Here, we demonstrate how iTransplant can be used to investigate clinicians' policies on organ offer acceptance from three perspectives:

> 1: Which criteria are important for clinicians' decisions on organ offers?
>
> 2: How does clinical practice change for patients with different characteristics?
>
> 3: What variation in practice exists between transplant centers?

Based on the similarity in organ acceptance rates and total numbers of organ offered, two transplant centers with ID codes 16864 and 19034 are selected for these investigative experiments. For convenience, we denote these centers as center A and center B, respectively. For the first two experiments, transplantation data from center A is used, while for the last one, we use iTransplant to learn separate policies for each center to explore the policy variation between the two centers.

**Discovering important match criteria**  The relative importance of different match criteria can be measured by the normalized policy signature $\tilde{\rho} = \rho/\|\rho\|_2$. With the distribution of normalized policy signature $\tilde{\rho}$ plotted in Figure 3, we are able to examine the learned policy at the population level. For most patients, as highlighted in Figure 3, the donor age and weight, presence of hepatitis B (HBV) or C (HBC), non-heart-beating donors, cause of donor death (donation after natural death), and high percentages of macrosteatosis (MaS) have negative contributions to the acceptance of organ offers. These observations are in line with risk factors reported in medical literature [33, 19] and guidelines [6].

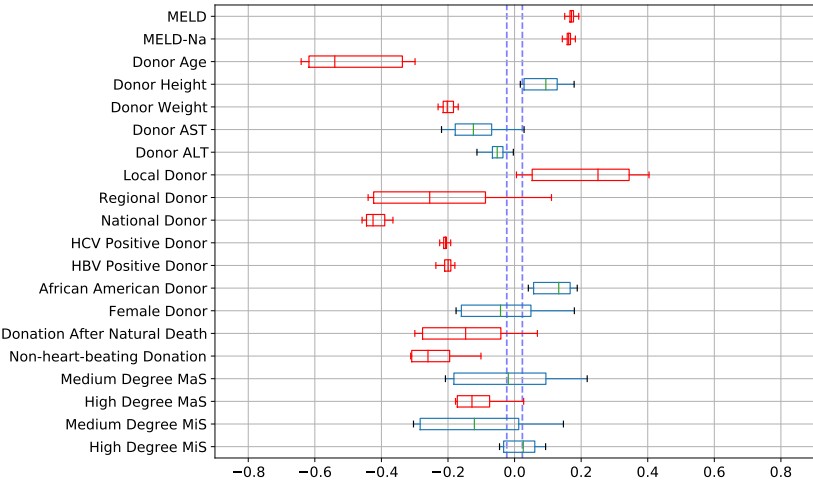

Figure 3: Distribution of normalized policy signature $\tilde{\rho}$ (x-axis) over match criteria (y-axis), see the Appendix for explanation. Criteria with weights between the dotted blue lines could be ignored with no more than 0.1% loss in the average precision score.

On the other hand, organs from local donors are preferred by clinicians while the organs shared at regional or national levels are more likely to be declined. This is supported by research on the impact of cold ischemia time (CIT), which is often a consequence of the organ origin [25, 9, 10]. Also,

Figure 3 shows that, for a given patient, a higher score from the model for end-stage liver disease (MELD) [44] will lead to higher chance of organ offer acceptance. As an indicator for 3-month mortality of hospitalized patients, a high MELD score suggests greater severity of the liver diseases, which may potentially encourage clinicians to accept the offered organ.

**Identifying patient-specific organ preferences of clinicians** Here, we use iTransplant to investigate the patient-specific organ preferences of clinicians and how their decision-making is affected by patient features. We cluster the patients in the test set for center A by applying the KMeans algorithm to the latent space $\mathcal{Z}$. The number of clusters is determined to be six such that the between-cluster variance of policy signatures $\rho$ is maximized. We find that patients in different clusters have significant deviations in policy signature indicating differences in clinical practice due to specific patient features. In Figure 4, we provide an overview of the policies for patients in cluster 1 and 2 displaying the deviation in policy with respect to the averaged policy signature $\bar{\rho}$ for all patients (see the Appendix for detailed discussion).

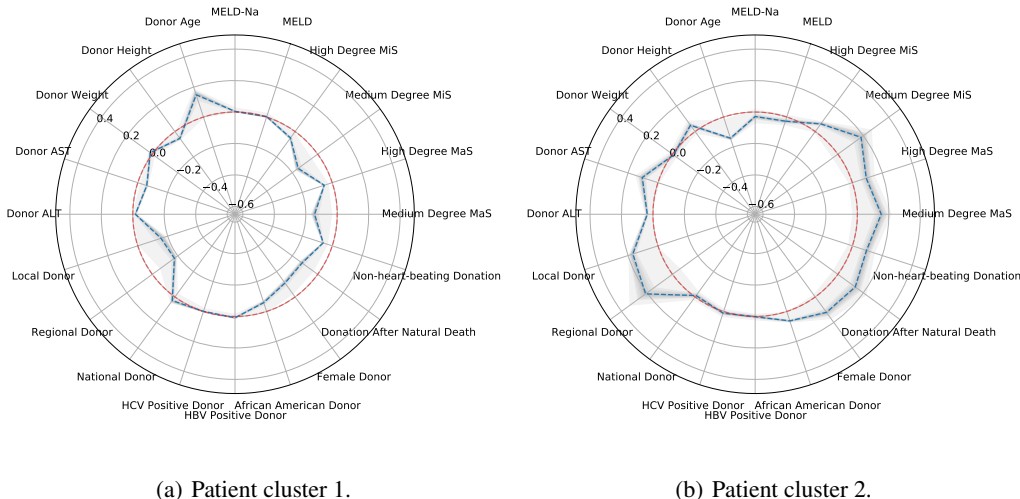

(a) Patient cluster 1.          (b) Patient cluster 2.

Figure 4: The deviations of policy signature $\Delta\rho = \rho - \bar{\rho}$ of two patient clusters.

Organs procured from regional donors are significantly more likely to be accepted for patients in cluster 2 than cluster 1 (Figure 4(a)). One significant difference between clusters 1 and 2 is whether patients have hepatocellular cancer (HCC). In cluster 2, 62.1% of patients are HCC positive with a MELD score below 20, while only 1.4% of patients in cluster 1 have both low MELD scores and the positive HCC status. Organs from regional donors typically have increased CIT compared to local donors. Prolonged CIT significantly reduces graft and patient survival, and thus organs from local donors are typically preferred (Figure 3). However, HCC positive patients with low MELD scores ($< 20$) are able to tolerate longer ischemia times without significant impact on graft survival [24], explaining the increased weight for regional donor (Figure 4(a)).

To illustrate the advantage and effectiveness of iTransplant in learning individualized policies for different patient groups, the policies learned by iTransplant and by logistic regression method are compared in the Appendix.

**Discovering variations in polices across transplant centers** Here, we illustrate the use of iTransplant as a tool for discovering patient-level policy variations across transplant centers. To achieve this, policies $\hat{\pi}^A$ and $\hat{\pi}^B$ are trained with organ offer data from center A and B, respectively, and are compared on the test set for center A. The probability of offer acceptance predicted by policies $\hat{\pi}^A$ and $\hat{\pi}^B$ was similar when averaged across all patients (difference: 0.005). However, there were significant differences on a per-patient basis, with a mean absolute error of $0.11 \pm 0.14$, resulting in different decisions for many patients. We identify a major divergence (predicted probabilities of offer acceptance are 0.72 and 0.21 for $\hat{\pi}^A$ and $\hat{\pi}^B$, respectively) of these two policies for an accepted organ offer associated with a 38 year old patient with high MELD score (39.0). The organ offered to this patient comes from a 53 year old donor with aspartate aminotransferase (AST) test value of

22.0, and the donor AST value and donor age were found to be two important factors related to such divergence in the considered policies. To demonstrate how this divergence in policies $\hat{\pi}^A$ and $\hat{\pi}^B$ affects the patient outcome, we assess the counterfactual impact of two donor characteristics on the organ offer acceptance probability by altering the age and AST test value of the donor (Figure 5).

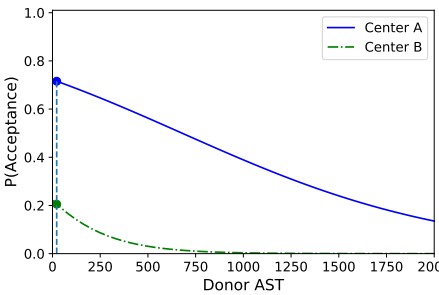 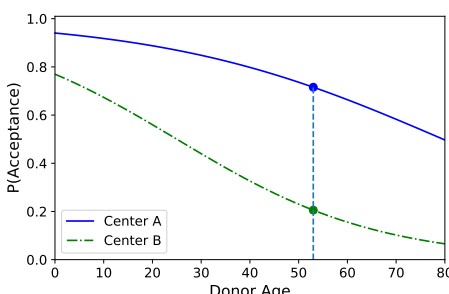

(a) Counterfactual impact of donor AST test value.  (b) Counterfactual impact of donor age.

Figure 5: Patient-level variations in policies from two different transplant centers.

The AST test value is an indirect measure of liver cell integrity, with a higher value in a donor suggesting more severe damage and therefore potentially worse post transplant outcome. Both policy $\hat{\pi}^A$ and $\hat{\pi}^B$ show preferences for organ offers with lower AST test value. However, for the considered patient, policy $\hat{\pi}^A$ is more tolerant to changes in donor AST test results. The low AST test result of the donor leads to a significantly higher likelihood of offer acceptance compared to policy $\hat{\pi}^B$ (Figure 5(a)). We find similar variations when examining the impact of donor age on the offer acceptance probability. While policy $\hat{\pi}^B$ has a high chance of accepting the organ offer from a young donor, the acceptance rate decreases steadily as the counterfactual donor age increases. For the policy $\hat{\pi}^A$, the acceptance rate is largely insensitive to the donor age until the age of the donor exceeds 30.

With the counterfactual impacts of donor features plotted in Figure 5, we can intuitively observe the variations in transplantation practices in the investigated transplant centers. This demonstrates the potential of iTransplant as a tool for clinicians to examine their own decision-making policies with existing transplantation data and the impact of changes to patient and donor characteristics.

## 6   Conclusion

In this paper, we propose an individualized transparent policy learning framework iTransplant to understand the organ offer related decision-making of clinicians. The individualized policy learned by iTransplant shows significant advantage compared to a global policy from logistic regression, highlighting the need to learn personalized policies. With real-world liver transplantation data, we conducted three investigative experiments demonstrating the use of iTransplant as a tool for discovering patient-level variations in transplantation practice and thus informing iterations of clinical decision support tools. While our experiments demonstrate that most decisions can be explained by clinical features recorded in observational datasets, it is likely that some decisions were made based on factors that were not considered. Future work could explore the impact of including additional match criteria and contextual information concerning unit activity and outcomes.

## Acknowledgements

This work was supported by Cystic Fibrosis Trust, Alzheimer's Research UK, the US Office of Naval Research (ONR), and the National Science Foundation (NSF, grant number 1722516). We thank the anonymous reviewers for their comments and suggestions. We thank the clinicians who participated in our survey. We thank Dr. Brent D. Ershoff for valuable discussions and counsel on the OPTN dataset and the observational features and decision criteria considered in this work. This work is based on the liver transplant data from OPTN, which was supported in part by Health Resources and Services Administration contract HHSH250-2019-00001C.

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
