# A   Appendix

## A.1   Data used for experiments

**Data source**   The experiments in our paper are conducted on a custom limited dataset (covering organ offers from January 1, 2003 to December 4, 2020) based on OPTN data as of December 4, 2020. The OPTN data are publicly available, and no patient identifiers or possibly identifiable information are included in the custom limited OPTN dataset used in this paper. Please see instructions on requesting OPTN data and the differences between custom limited datasets and patient-identified datasets on https://optn.transplant.hrsa.gov/data/request-data/data-request-instructions/.

**Selection of organ offer data**   In the OPTN system, when an organ offer is assigned to a potential recipient in one transplant center, there are two types of initial response – *provisional acceptance* and *rejection*. More details of the donor organ are then provided to centers of *provisional acceptance*, and the organ will be offered to one of these centers based on their final decisions (*acceptance* or *rejection*) and the priority of their patients in the organ offering system. Detailed OPTN policies for organ allocation and organ offer acceptance can be found in https://optn.transplant.hrsa.gov/governance/policies/. The reason of rejection was provided if the organ offer was declined by a transplant center, while no reason was recorded for *acceptance* and *provisional acceptance*. In some cases, the initial response of *provisional acceptance* is not updated to *acceptance* or *rejection*, and the final decision is not known. Note that in our custom limited dataset from OPTN, with 0.96% of *acceptance* and 95.58% of *rejection*, only 3.46% of responses to organ offers are *provisional acceptance*. Thus, in our experiment, we only include organ offers with a final decision of *acceptance* or *rejection*. In some special cases, the reasons for organ offer rejection are either administrative in nature (e.g., "surgeon unavailable" and "heavy workload") and do not directly relate to possible patient or organ factors available in our datasets (e.g., "patient's condition improved, transplant not needed"). For the experiments in this paper, we select rejected organ offers of twelve cases, covering 87.7% of all rejected offers. These considered reasons of rejection include: donor age or quality (code 830, 922), donor size or weight (code 831, 923), organ-specific donor issue (code 837), distance to travel or ship (code 824), donor quality (code 921), positive serological tests of donor (code 834), donor organ anatomical damage or defect (code 836), donor blood type (code 832, 924) and abnormal biopsy of donor liver (code 942).

**Patient and donor organ features used in experiments**   Based on domain knowledge on liver transplantation, subsets of patient and donor organ features related to decisions on organ offers are selected to construct the patient feature space $\mathcal{X}$ and organ feature space $\mathcal{O}$, respectively. The patient feature vector $\mathbf{X} \in \mathcal{X}$ and organ feature vector $\mathbf{O} \in \mathcal{O}$ are calculated using the original transplantation data. Organ offers which contain missing values in patient or organ features were excluded from the dataset.

The list of patient features selected includes: age, gender, height, weight, blood type, body mass index (BMI), creatinine concentration, international normalized ratio (INR), bilirubin concentration, sodium concentration (Na), MELD score, albumin concentration, status of ascites, ethnicity, status of dialysis, status 1a, status of hepatocellular carcinoma (HCC), indicator of hepatocellular carcinoma exception point, functional status of patient, indicator of life support status, indicator of whether the patient was on a mechanical ventilator, status of portal vein thrombosis, history of prior abdominal surgery, number of previous transplants and primary cause of liver disease.

The list of donor organ features selected includes: age, gender, height, weight, blood type, creatinine concentration, bilirubin concentration, aspartate aminotransferase (AST) test result, alanine aminotransferase (ALT) test result, indicator of non-heart beating donor, cause of death for deceased donor, ethnicity, donor organ origin, death mechanism of donor, circumstances of death (natural death or not), organ procure type (whole or split), percentage of macrosteatosis (MaS), percentage of microsteatosis (MiS), hepatitis B serostatus and hepatitis C serostatus.

For the experiments in this paper, we only consider adult patients that joined the waitlist after 2002. We also removed patients who have been retransplanted, cases of living donor transplantation, and multiorgan transplantation. Thereafter, records with missing values for selected patient and donor organ features are removed.

**Match criteria used in experiments**   Most match criteria used in the experiments are directly obtained as donor organ features. Here, we provide additional explanations for some of match criteria related to patient features. The donor organ origin (local/regional/national donor) with respect to patient location is calculated based on distance information in the patient-organ match classification from OPTN data. According to [21, 44], for the MELD and MELD-Na score calculation, we use the following:

$$\text{MELD} = \lfloor 6.43 + 9.57 \times \log(\text{creatinine}) + 3.78 \times \log(\text{bilirubin}) + 11.2 \times \log(\text{INR}) \rceil,$$

$$\text{MELD-Na} = \begin{cases} \text{MELD}, & \text{if MELD} < 12, \\ \text{MELD} + 1.32 \times (137 - \text{Na}) - 0.033 \times (137 - \text{Na}) \times \text{MELD}, & \text{otherwise,} \end{cases}$$

where $\lfloor \cdot \rceil$ rounds its input to the nearest integer. In addition, the creatinine value is clamped into $[1, 4]$ and is adjusted to $4.0$ if the patient had dialysis at least twice in the past week. The INR and bilirubin values are clamped into $[1, 1 \times 10^{10}]$ to avoid negative MELD scores. For MELD-Na score, the Na value is adjusted to the range of $[125, 137]$, and the final MELD-Na score is clamped into range $[6, 40]$.

## A.2   Additional explanations for the network structure of iTransplant

**Reconstruction loss**   For the auto-encoder part in iTransplant, the standard reconstruction loss $\mathcal{L}_{Reconstruction}$ is adopted to guide the learning of latent representation of patient features. Here, $\mathcal{L}_{Reconstruction}$ is calculated as the mean squared error between the reconstructed patient feature $\hat{\mathbf{X}}$ from the decoder network and the original patient feature vector $\mathbf{X} \in \mathcal{X}$ as shown in equation (3)

$$\mathcal{L}_{Reconstruction} = \int_{\mathcal{X}} \|\mathbf{X} - \hat{\mathbf{X}}\|_2^2 d\mathbf{X}. \tag{3}$$

**Joint loss function**   The three loss functions of policy loss $\mathcal{L}_{Policy}$, reconstruction loss $\mathcal{L}_{Reconstruction}$ and consistency loss $\mathcal{L}_{Consistency}$ are added together for the training of iTransplant. Note that the gradients from $\mathcal{L}_{Consistency}$ only apply to the policy selector network while gradients from $\mathcal{L}_{Policy}$ are passed to the encoder network to guide the representation learning of patient features in the latent space $\mathcal{Z}$ together with $\mathcal{L}_{Reconstruction}$.

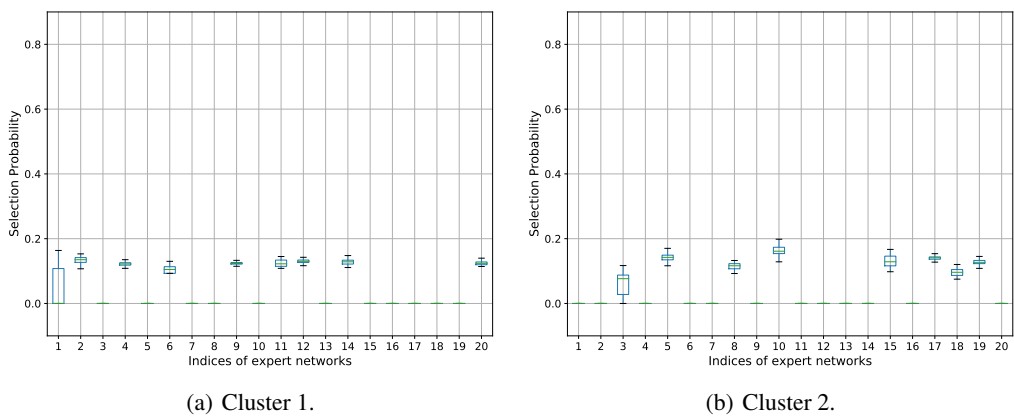

(a) Cluster 1.

(b) Cluster 2.

Figure 6: The selection probability of expert networks for patients in two different clusters.

**Mixture of expert (MoE) layer**   The MoE layer of the policy selector network in iTransplant contains a gating network and $K$ expert networks. In this paper, the gating network is represented by a $K$-by-$m$ matrix $\mathbf{G}$, where $m$ is the dimension of the latent space $\mathcal{Z}$. The expert networks are ranked based on the selection likelihoods calculated from the matrix product $\mathbf{GZ}$, where $\mathbf{Z} \in \mathcal{Z}$ is the latent representation of patient features $\mathbf{X}$, and outputs of the top-$k$ experts in the MoE layer are combined to make predictions. For patients from different cohorts, variations in their latent representations in $\mathcal{Z}$ would lead to the selection of different expert networks. The distribution of expert selection

probability with organ offer data from transplant center with ID code 16864 is illustrated in Figure 6. With $K = 20$ experts available, the gating network in MoE layer tends to assign different sets of expert networks for patients from different clusters. Note that the patient clusters here are the same with those in the main manuscript.

## A.3 Experiment setup

**Selection of transplant centers** For experiments in this paper, organ offer data from ten transplant centers with sufficient number (over 1,000) of accepted offers after removal of missing data are selected. Some basic information of these ten centers can be found in Table 4.

Table 4: Statistics of organ offers in the selected centers.

| CENTER ID CODE | ORGAN OFFERS[2] | ACCEPTANCE RATE | ACCEPTED ORGAN OFFERS |
|---|---|---|---|
| 20522 | 12140 | 18.59% | 2257 |
| 19034 | 17608 | 11.36% | 2001 |
| 6820 | 13120 | 12.90% | 1693 |
| 7471 | 16429 | 10.29% | 1691 |
| 23312 | 31013 | 5.18% | 1607 |
| 14942 | 11148 | 13.37% | 1491 |
| 16864 | 14030 | 10.63% | 1491 |
| 23808 | 29964 | 4.86% | 1457 |
| 25110 | 31313 | 4.38% | 1371 |
| 13609 | 13764 | 9.63% | 1326 |

**Model configurations** In the main manuscript, baselines of (per-cluster) logistic regression, (per-cluster) decision tree, locally weighted regression (LOWESS), random forest, global variable selection with LASSO, instance-wise variable selection with INVASE and a neural network-based behavioural cloning (BC) model are used for comparison with iTransplant. For the baselines of logistic regression, decision tree, random forest and LASSO, implementations in scikit-learn[3] are adopted. The logistic regression-based policies in iTransplant include no interception terms, and no penalty is applied to the policy signatures identified by iTransplant. In line with this, we set zero-penalty and zero-interception options for the coefficients in logistic regression. For the baselines of per-cluster logistic regression and per-cluster decision tree, KMeans algorithm is applied to discover patient clusters. Logistic regression models and decision trees are then trained independently for each of these clusters. For BC, a multi-layer perceptron (MLP) is used to model the map from an organ offer $\mathbf{s}$ to the probability of offer acceptance. For INVASE, three MLPs are utilized as the baseline, critic and selector networks, respectively. Similarly, in iTransplant, the encoder and decoder networks and the experts in the MoE layer are all MLPs with linear activation functions for outputs. The activation functions for non-output layers in BC, INVASE and iTransplant are set to be *LeakyReLU*.

**Model inputs** For all baselines, match criteria $\mathcal{T}(\mathbf{s})$ in the criteria space $\mathcal{M}$ are used by the models to achieve action matching with the expert policy. The patient feature vector $\mathbf{X}$ is provided additionally to per-cluster logistic regression and per-cluster decision tree models for the clustering of patients via KMeans. Similarly, the patient feature vector $\mathbf{X}$ is also provided to LOWESS for the measurement of patient similarity. To provide the upper bounds of performance, both patient features space $\mathcal{X}$ and the criteria space $\mathcal{M}$ are used as inputs to BC. Further evaluation with respect to different model inputs for the baselines can be found in Section A.4.

**Selection of hyperparameters** For hyperparameter selection, benchmark and investigative experiments, the learning rate for all neural networks is set to be $1 \times 10^{-3}$ and the maximum number of training iterations is set as 200. The early stopping technique is adopted for BC and iTransplant in the training phase to avoid over-fitting.

For MLPs in BC, INVASE and iTransplant, the number of hidden layers and the number of units in each layer are denoted as $l_n$ and $h_n$, respectively. For INVASE, the hyperparameter $\lambda$ refers to the

---

[2]Organ offers with missing values in selected patient and organ features are not counted.
[3]https://scikit-learn.org/stable/

coefficient for $l_1$ penalty term for its instance-wise feature selection masks. For iTransplant, the multiplier for load-balancing in the MoE layer is kept as is in the implementation[4] (loss_coef=0.01). The number of experts in the MoE layer, the number of selected experts for policy signature identification and coefficient for the consistency loss are denoted as $K$, $k$ and $\lambda$, respectively.

For (per-cluster) logistic regression and decision tree models and random forest, the number of clusters, number of decision trees and maximum depth of decision trees are denoted as $n_{cluster}$, $n_{tree}$ and $h_{depth}$, respectively. For LASSO, the $l_1$ regularizer coefficient is denoted with $\alpha$. The bandwidth of the LOWESS algorithm is denoted as $\tau$.

The range of hyperparameters considered for each method mentioned above are given as follows.

- Logistic Regression: $n_{cluster} \in \{1, 2, 4, 8\}$.
- Decision tree: $n_{cluster} \in \{1, 2, 4, 8\}, h_{depth} \in \{5, 10, 15\}$.
- LASSO: $\alpha \in \{0.01, 0.1, 1.0, 5.0\}$.
- Random Forest: $n_{tree} \in \{10, 50, 100, 150\}, h_{depth} \in \{5, 10, 15\}$.
- LOWESS: $\tau \in \{0.01, 0.1, 1, 10\}$.
- BC: $h_n \in \{10, 20, 30, 40, 50\}, l_n \in \{2, 4, 6, 8\}$.
- INVASE: $h_n \in \{20, 30\}, l_n \in \{2, 4, 6\}, \lambda \in \{0.01, 0.1, 0.5\}$.
- iTransplant: $h_n \in \{30, 40, 50\}, \lambda \in \{0.01, 0.05\}, K \in \{10, 20\}, k \in \{4, 8\}$.

These hyperparameters are selected via a grid search with five-fold cross-validation on organ offer data from transplant center with ID code 20522 as listed in Table 4. The hyperparameter selection results are given as follows.

- Logistic Regression: $n_{cluster} = 1$.
- Per-cluster Logistic Regression: $n_{cluster} = 2$.
- Decision tree: $n_{cluster} = 1, h_{depth} = 5$.
- Per-cluster Decision tree: $n_{cluster} = 2, h_{depth} = 5$.
- LASSO: $\alpha = 0.01$.
- Random Forest: $n_{tree} = 150, h_{depth} = 15$.
- LOWESS: $\tau = 1$.
- BC: $h_n = 50, l_n = 4$.
- INVASE: $h_n = 30, l_n = 2, \lambda = 0.01$.
- iTransplant: $h_n = 30, \lambda = 0.01, K = 20, k = 8$.

**Data split and random seeds** For the benchmark and investigative experiments, organ offer data from each transplant center are split to training and test sets with the ratio of $8 : 2$. For the purpose of early stopping of iTransplant and BC, 20% of organ offers are randomly extracted from the training set to evaluate the action matching performance during training process. In the benchmark, random seed for data split is set to be 40108642, and the models are initialized with ten different integers generated with the random seed of 19260817 to calculate error bars.

**Computing resources and environment** All of the experiment results in this paper are obtained on a CPU of Intel Core™ i5-10210U with maximum RAM of 16 GB. The Python environment is set up under the Windows Subsystem for Linux (WSL 2) with Debian GNU/Linux 10 (buster). The Python version is 3.8.7 with the cpu version of torch 1.9.0. Detailed information can be found in our public code repositories (https://github.com/yvchao/iTransplant and https://github.com/vanderschaarlab/iTransplant).

---

[4] https://github.com/davidmrau/mixture-of-experts

### A.4 Supplementary experiment results

**Contribution of additional donor organ features**   As complementary evidence to the benchmark results reported in the main manuscript, the performances of LASSO, INVASE, BC with different input spaces are compared with iTransplant in Table 5. No significant changes in performance are observed from the benchmark results of BC and INVASE when the match criteria space $\mathcal{M}$ constructed used by iTransplant is replaced by donor organ feature space $\mathcal{O}$. According to Table 5, with full set of organ feature variables in input space $\mathcal{X} \times \mathcal{O}$, the prediction performances of LASSO are largely improved. However, iTransplant still significantly outperforms LASSO even with a subset of donor organ features in the criteria space $\mathcal{M}$.

Table 5: Benchmark results with different input space.

| METHOD | INPUT SPACE | AUC-ROC | AUC-PRC | LL |
|---|---|---|---|---|
| LASSO | $\mathcal{M}$ | 0.777±0.064 | 0.314±0.068 | -0.570±0.045 |
| LASSO | $\mathcal{X} \times \mathcal{M}$ | 0.875±0.053 | 0.439±0.066 | -0.482±0.049 |
| LASSO | $\mathcal{X} \times \mathcal{O}$ | 0.877±0.051 | 0.443±0.064 | -0.478±0.048 |
| INVASE | $\mathcal{M}$ | 0.790±0.062 | 0.341±0.071 | -0.541±0.064 |
| INVASE | $\mathcal{X} \times \mathcal{M}$ | 0.875±0.053 | 0.439±0.066 | -0.482±0.049 |
| INVASE | $\mathcal{X} \times \mathcal{O}$ | 0.877±0.051 | 0.443±0.064 | -0.478±0.048 |
| BEHAVIORAL CLONING | $\mathcal{M}$ | 0.806±0.059 | 0.361±0.073 | -0.515±0.072 |
| BEHAVIORAL CLONING | $\mathcal{X} \times \mathcal{M}$ | 0.899±0.043 | 0.502±0.067 | -0.383±0.067 |
| BEHAVIORAL CLONING | $\mathcal{X} \times \mathcal{O}$ | 0.899±0.042 | 0.505±0.067 | -0.390±0.070 |
| ITRANSPLANT (OURS) | $\mathcal{X} \times \mathcal{M}$ | 0.895±0.045 | 0.502±0.062 | -0.396±0.069 |
| ITRANSPLANT (OURS) | $\mathcal{X} \times \mathcal{O}$ | 0.898±0.048 | 0.508±0.064 | -0.385±0.076 |

**Comparison with per-cluster logistic regression with additional interaction terms**   As shown in Table 6, although including additional pair-wise interaction terms between the match criteria considered in space $\mathcal{M}$ can help to improve the performance of per-cluster logistic regression model, there is still a large gap to the action matching performance achieved by iTransplant. Although the prediction performance of a (per-cluster) logistic regression model is improved slightly with additional interaction terms, a significant gap in performance to iTransplant remains. In comparison, our proposed method, iTransplant, achieves similar performance to the black-box BC model without any interaction terms. Here, we highlight that our method allows clinicians to choose the match criteria considered in space $\mathcal{M}$ based on their expertise, and additional interaction terms can be easily included in the criteria space $\mathcal{M}$.

Table 6: Benchmark with per-cluster logistic regression with interaction terms.

| METHOD | AUC-ROC | AUC-PRC | LL |
|---|---|---|---|
| LOGISTIC REGRESSION (LR) | 0.794±0.054 | 0.341±0.061 | -0.538±0.051 |
| PER-CLUSTER LR | 0.803±0.049 | 0.352±0.063 | -0.527±0.048 |
| PER-CLUSTER LR (WITH INTERACTION TERMS) | 0.824±0.054 | 0.371±0.073 | -0.490±0.063 |
| ITRANSPLANT (OURS) | 0.898±0.048 | 0.508±0.064 | -0.385±0.076 |

**Average performance on individual centers**   In addition to the benchmark result in our main manuscript, the average AUC-ROC and AUC-PRC scores of BC, INVASE, per-cluster decision tree, per-cluster logistic regression and iTransplant calculated on a per-center basis are plotted in Figure 7. In all centers, iTransplant effectively reduces the performance gap with (and in some cases surpasses) black-box models (BC and INVASE) while significantly outperforming the white-box models of logistic regression and decision tree, even when compared on a per-cluster basis. This demonstrates the benefit of identifying policy aligned personalized decision policies compared to learning policy-agnostic cluster-wise policies.

**Details of consistency evaluation**   For the experiment on consistency, we focus on the twenty criteria in the criteria space $\mathcal{M}$. For perturbations to donor organ features in criteria space, iTransplant

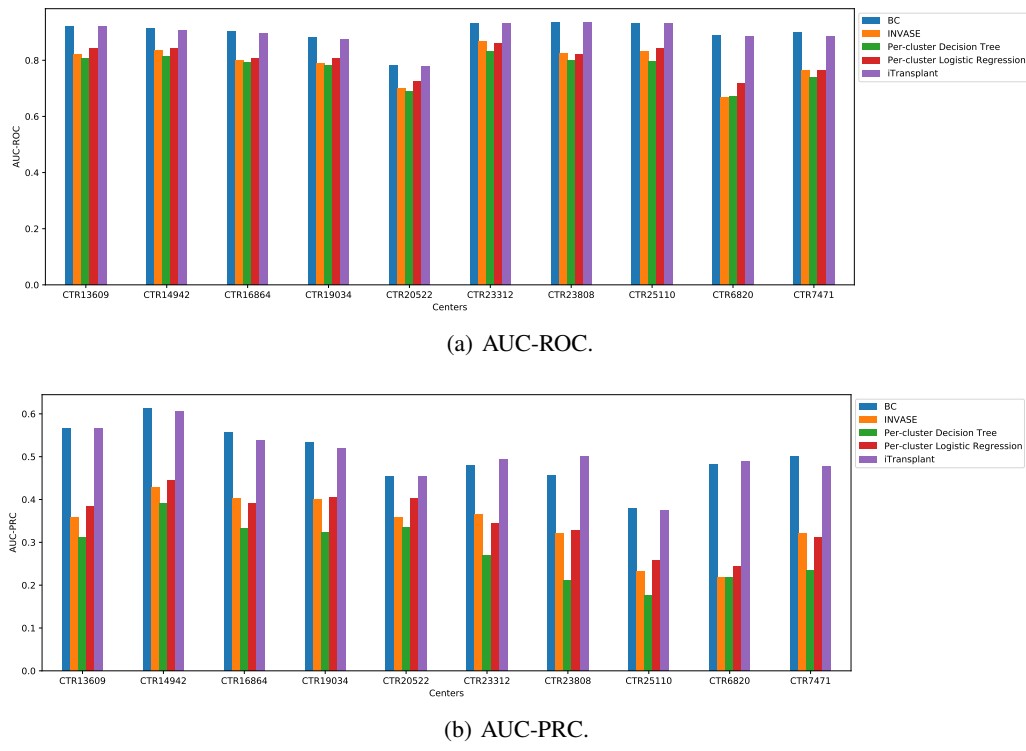

(a) AUC-ROC.

(b) AUC-PRC.

Figure 7: Average performance on organ offer data from ten selected centers.

is innately consistent with its predictions. In the meantime, when patient features related to MELD and MELD-Na scores changes, iTransplant could be inconsistent due to the corresponding deviations of policy signature $\rho$. As mentioned in the main manuscript, we measure the average consistency score over all twenty match criteria, and perturbed samples leading to no changes in match criteria are ignored. The consistency scores of considered methods on 200 organ offers randomly selected from test set of transplant center with ID code 20522 and the average AUC-ROC and AUC-PRC scores on all ten centers are reported in Table 7. According to the consistency scores, the post-hoc interpretations by applying LIME to BC and INVASE provide almost no useful insights under moderate perturbations to input features. In contrast, the white-box part of iTransplant contributes hugely to the consistency score. As reported in Table 7, The consistency of iTransplant can be further improved via a strong

Table 7: Consistency evaluation of interpretations.

| METHOD | CONSISTENCY |
|---|---|
| BC | 46.8% (WITH LIME) |
| INVASE | 41.7% (WITH LIME) |
| iTRANSPLANT, $\lambda = 0.0$ | 91.3% |
| iTRANSPLANT, $\lambda = 0.01$ | 85.1% |
| iTRANSPLANT, $\lambda = 0.1$ | 99.7% |

consistency regularization ($\lambda = 0.1$). However, due to the complex interactions between the policy loss, reconstruction loss and the consistency loss, smaller consistency regularization ($\lambda = 0.01$) may not always lead to an increase in consistency. In practical applications, the trade-offs between consistency and the action-matching performance should be carefully considered.

**Performance gain of iTransplant over logistic regression** As discussed in the main manuscript, the patients are clustered into six different clusters based on their latent representation in space $\mathcal{Z}$. With AUC-PRC score as the performance metric, detailed comparison of the prediction performance

of iTransplant and logistic regression in each of these patient clusters is shown in Table 8. We can find that the proposed method iTransplant achieves significant performance gains in all six clusters.

Table 8: Performance (AUC-PRC) gain over logistic regression in each patient cluster.

| CLUSTER | SAMPLE NUMBER | ITRANSPLANT | LOGISTIC REGRESSION |
|---------|---------------|-------------|---------------------|
| 1 | 588 | **0.152** | 0.133 |
| 2 | 311 | **0.449** | 0.406 |
| 3 | 725 | **0.551** | 0.514 |
| 4 | 614 | **0.032** | 0.025 |
| 5 | 522 | **0.654** | 0.576 |
| 6 | 46 | **0.411** | 0.187 |

**Policy deviations in each patient clusters** As supplementary results of the main manuscript, the policy deviations in the first three patient clusters with respect to the averaged policy $\hat{\pi}_{\bar{\rho}}$ are plotted in Figure 8. It can be found that iTransplant is able to identify personalized organ offer acceptance policies for each patient group, and we argue that this is one of the main reasons for the performance gains observed in Table 8.

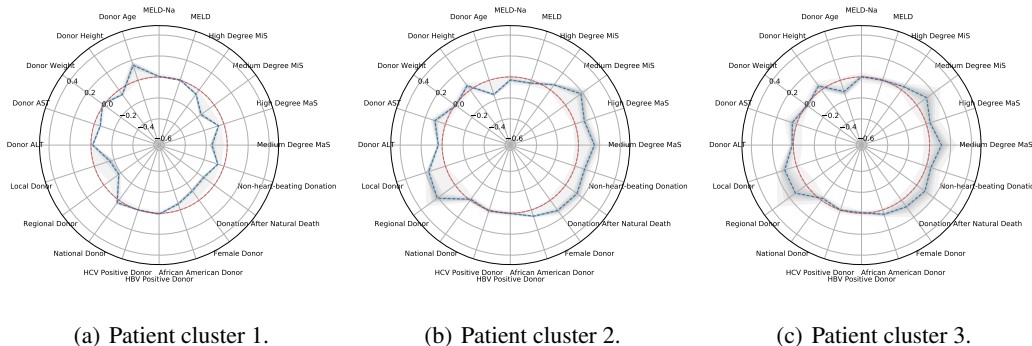

(a) Patient cluster 1.  (b) Patient cluster 2.  (c) Patient cluster 3.

Figure 8: Deviations of policy signature $\Delta\rho = \rho - \bar{\rho}$ in all three clusters.

**Statistics for key patient features in each cluster** Statistics about the distribution of two key patient features (MELD score and HCC status) in different clusters are given in Table 9.

Table 9: Key patient feature distributions in each patient cluster.

| CLUSTER | MELD SCORE | | | | | HCC STATUS | |
|---------|------------|--------|--------|--------|-----------|------------|----------|
| | $[0, 10)$ | $[10, 20)$ | $[20, 30)$ | $[30, 40)$ | $[40, +\infty)$ | POSITIVE | NEGATIVE |
| 1 | 14.8 % | 76.36% | 8.67% | 0.17% | 0.00% | 1.53% | 98.47 % |
| 2 | 47.59% | 51.13% | 1.29% | 0.00% | 0.00% | 62.38% | 37.62% |
| 3 | 0.97% | 65.52% | 27.59% | 4.41% | 1.52% | 0.00% | 100.00% |
| 4 | 21.82% | 67.92% | 9.45% | 0.81% | 0.00% | 0.33% | 99.67% |
| 5 | 3.26% | 51.34% | 33.14% | 9.00% | 3.26% | 0.00% | 100.00% |
| 6 | 0.00% | 2.17% | 84.78% | 6.52% | 6.52% | 0.00% | 100.00% |

**Decision boundaries in all patient clusters** Together with the policy deviations in each cluster, decision boundaries of the in-cluster average policies identified by iTransplant are plotted in Figure 9 (only clusters with no less than 30 accepted organ offers are considered). We can find that iTransplant can properly separate the positive and negative samples with the personalized policies identified in each cluster. We further demonstrate the benefit of personalized policy learning with the comparison of decision boundaries provided by a logistic regression model and iTransplant for patient cluster

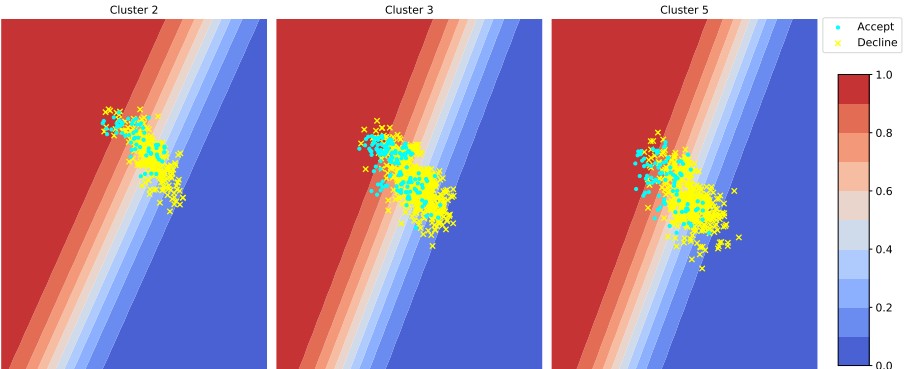

Figure 9: Decision boundaries in considered clusters. The probabilities of organ offer acceptance given by iTransplant are plotted in the background with colors of red and blue for high and low likelihood of offer acceptance, respectively. Organ offers for each patient clusters are projected to the hyperplanes determined by the averaged policy signatures $\rho$ of the policies given by logistic regression and iTransplant in policy space $\Pi$. For each cluster, declined organ offers are marked with yellow crosses while accepted offers are marked with cyan dots.

2 in Figure 10. The optimal decision boundary for positive and negative samples in the projection hyperplane is very close to the one from iTransplant while a logistic regression model is unable to provide customized decision boundaries for patients in different subgroups, which explains the performance gaps observed in Table 8.

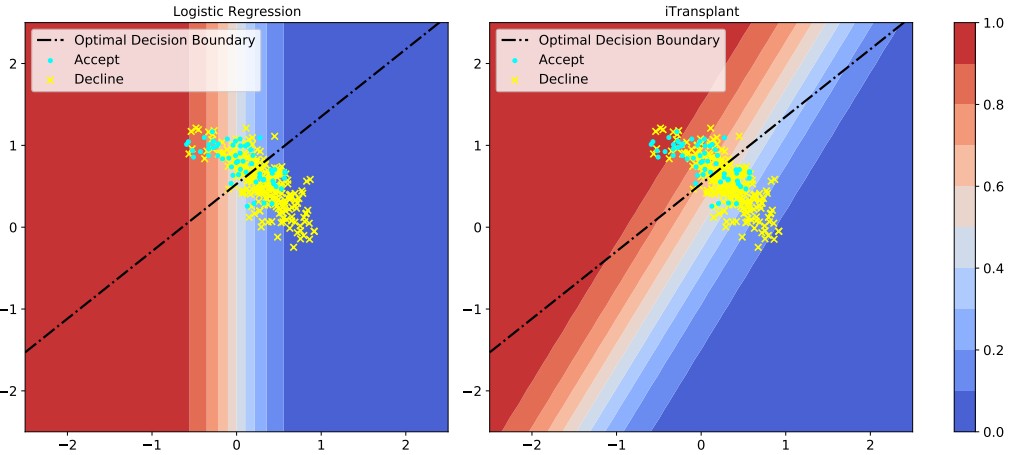

Figure 10: Decision boundaries in patient cluster 2. (L) Logistic regression, (R) iTransplant. The optimal decision boundary in the projection hyperplane is obtained via the support vector machine algorithm and is plotted as black dotted lines.

## A.5 Further discussion on consistency

**Importance of consistency** Consistency is proposed as one of the key desiderata of practical inverse decision-making approaches in this paper. For the insights from inverse decision-making methods to be helpful for decision makers, the interpretations need to be consistent under moderate perturbations in input space. Practically, this means that if a feature has a positive weight, a moderate increase to the value of the feature should increase the probability of acceptance (and for a binary feature with positive weight, flipping the feature should reduce the probability of acceptance). Without this

property, counterfactual analysis cannot be reliably performed, limiting the value of any interpretation or understanding gained from inverse decision-making.

**Consistency and non-smooth preferences over patient features**  Clinicians' preferences over donor organs could be non-smooth functions over patient features, which may lead to potential failure of methods (like logistic regression) that issues extreme consistent interpretations to clinicians' decisions. In this paper, we requires the consistency between latent space representations of patient features and policies, rather than consistency directly between the patient features and policies. Thus, despite having a non-zero consistency regularization term defined in equation (2) of our main manuscript, altering patient features could still drastically change the latent representation, thus changing the policy significantly, which is the desired outcome in our method. In addition, we note that the logit model in equation (1) allows a single criterion in the criteria space $\mathcal{M}$ to flip the decision and thus our modeling framework is able to sufficiently accommodate the non-smooth preferences scenarios.

### A.6  A short survey on interpretability of machine learning methods

We have conducted a short survey with six clinicians in an attempt to further validate the usefulness of our method. After a brief introduction to decision tree, logistic regression and neural network models, four questions as shown in Figure 11 were shown to each clinician individually, and the surveyed clinicians are asked to give their feedback based on their own understanding of these questions. The survey was conducted blindly, i.e., the clinicians were given no context as pertains to our proposed method.

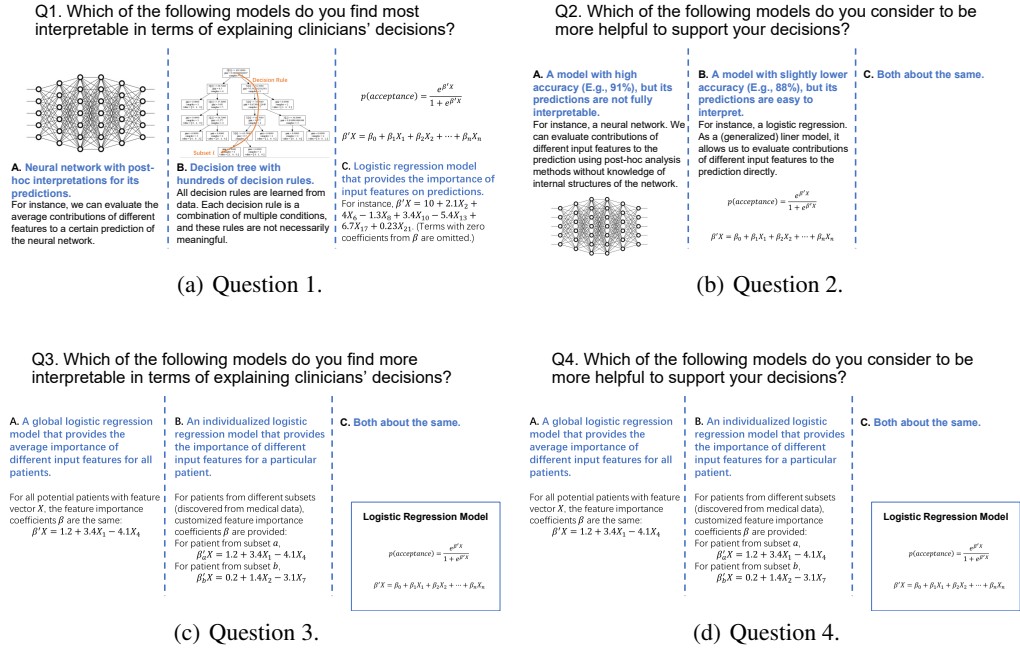

Figure 11: Four questions in the small survey on interpretability.

The results of our survey are shown in Table 10. Four out of the six surveyed clinicians consider logistic regression to be more or equally helpful compared to a high-precision black-box model. In addition, only one surveyed clinician considers the post-hoc interpretations of neural network-based models to be more interpretable than white-box models like decision tree and logistic regression. Notably, all six clinicians deemed individualized logistic regression models more informative and practical than a general logistic regression model. Despite the relatively small sample size, we believe that these results provide evidence of the importance of interpretability in our proposed approach and

the usefulness of our method compared to alternate forms of interpretability, including basic logistic regression models.

Table 10: Responses from six surveyed clinicians.

| CLINICIAN | QUESTION 1 | QUESTION 2 | QUESTION 3 | QUESTION 4 |
|---|---|---|---|---|
| CLINICIAN 1 | B,C | B | B | B |
| CLINICIAN 2 | B | B | B | B |
| CLINICIAN 3 | C | B | B | B |
| CLINICIAN 4 | B,C | C | B | B |
| CLINICIAN 5 | A | A | B | B |
| CLINICIAN 6 | B | A | B | B |

## A.7 Limitations of our work

While our experiments demonstrate that most decisions can be effectively explained by iTransplant with clinical features recorded in observational datasets, some decisions were still likely to be made based on factors that were not considered in this paper. It is therefore important to explore the impact of including additional match criteria and contextual information concerning unit activity and outcomes in iTransplant for practical applications. In the meantime, the performance gains of iTransplant comes from the black-box policy selector model. Although the policies identified for individual patients are fully human comprehensible, for real-world applications, it is necessary to build the trust of clinicians on the black-box part of iTransplant via properly designed mechanisms for human-machine interactions.

We would like to discuss some additional relevant limitations as follows.

**Misinterpretation of correlations** The insights provided by our method could be used to positively impact future organ transplant practices. However, we need to highlight that the decision drivers discovered by iTransplant only signify potential correlations between certain criteria and real decisions on organ offers. Such correlations between features in the criteria space and decisions identified by our method should not be interpreted as causal, and care must be taken when interpreting the output of any inverse decision-making approach, including our method.

**Impact from organ offering policies** Currently, specific organs are offered to clinicians for a specific recipient, with that offer being predicated on nationally agreed guidelines. A clinician will accept or reject the offer based on their knowledge, experience and discussion with the recipient. Nevertheless, transplant centers have very variable acceptance rates of offered organs, with some consistently accepting a higher percentage than others. Hence, indeed, the organ offering policy determines which organ offers clinicians must make decisions on, which ultimately shapes the observed distribution of organ offers in the data. However, the goal of our method is precisely to understand the observed decisions of clinicians—i.e., given the organ allocation policy in the real transplantation system.

To this end, we seek interpretable parameterizations of the drivers of human decisions that are predictive of these observed decisions. Of course, these parameterizations are only valid for the observed data (i.e., "in-distribution" prediction), and should not be used in scenarios where the organ offering policy is significantly different (i.e., "out-of-distribution" prediction). In this sense, this caveat is no different from any inverse decision-making or supervised learning setting—a change in domain will generally degrade the analysis.

**No hidden confounders** In our method, it is assumed that drivers of observed decisions in the dataset are fully observable or can be inferred from observable features in the dataset. However, as a limitation of many inverse decision-making approaches, it is likely that some decisions were made based on factors that were not included in the observational dataset, which may bias the insights discovered by our method and lead to lower prediction performance.

**Confirmation bias** Any endeavor aimed at "pattern discover" will run the risk of confirmation bias. In most realistic use cases, there will be no absolute "ground truth" available for validation, and

our inverse decision-making approach is not immune to this limitation. Importantly, however, we emphasize that we are not in the business of performing statistical inference or testing (whence the formal quantification of "statistical significance" would require rigid procedures to ensure validity). To the contrary, we are engaged in the orthogonal mission of generating interpretable hypotheses on the underlying drivers of human decisions—in particular, drivers that are highly predictive of observed decisions. The purpose of the investigative experiments is to illustrate that our method is capable of proposing hypothetical decision policies that could be matched with real clinical findings or guidelines, which makes our method useful for clinicians to review and improve their decisions. In the meantime, for practical applications of our method, validation and additional analysis by domain experts is necessary to validate the discovered insights.

**Linearity in the criteria space**  The assumption of linearity in the criteria space (or the reward structure) is common in inverse decision-making literature that aims to identify interpretable decision policies from the observable decision history of an agent. We take the same assumption of linearity to obtain human comprehensible decision policies. However, the validity of this assumption strongly depends on the selection of feature maps (criteria) in the criteria space. Misspecification of the criteria space may lead to misleading interpretations and poor performance of inverse decision-making methods. Thus, careful validation by domain experts is necessary for practical applications.

**Greedy decision policy**  The OPTN database describes donor organ-recipient pairs, where the organ was offered according to national guidelines to an individual recipient, and accepted or declined by a clinician. The decision to accept or decline was not informed by knowledge of the whole waitlist of potential recipients nor the future availability of donor organs. In line with this, we take the assumption that the decision sequences of clinicians are greedy and not purposeful (accounting for outcomes of future decisions). The assumption of greedy decision policy is necessary for modeling the organ offer acceptance as a one-step decision-making problem and was deemed reasonable in our discussions with clinicians. However, this assumption blocks the application of our approach on sequential decision-making settings where the current decision is affected by potential outcomes at future states. We leave the extension of iTransplant to such settings for our future work.

**Missing data**  Missing data is an important issue for practical applications of machine learning. That said, our proposed method tackles a problem orthogonal to that of missing data. To be clear, neither our method nor any of the included benchmarks have any "built-in" capabilities designed for missing data. However, in practice our method (and benchmarks) is compatible with any existing data imputation algorithms.

## A.8   Potential applications

As mentioned in the main manuscript and the Appendix, our model is proposed as a tool to help clinicians to review and identify potential drivers of their clinical decisions. As described in the introduction, clinical decision-making in organ transplantation is poorly understood with a high proportion of organ offers declined and substantial variation in clinical practice. The intended use of a tool such as iTransplant would be to stand beside the clinician decision-makers so that they can reflect upon and better understand the drivers for their decision. These issues are not unique to liver transplantation, and variation in practice has been studied across medicine [2, 12], including cancer [42, 37] and intensive care [11].

As illustrated in our experiments, our method can be used to study and gain greater understanding of these clinical phenomena, and the insights obtained via our method could ultimately be used to improve future decisions. It is worth noting that the criteria space in our method is not limited to the one discussed in our paper. In fact, clinicians could use iTransplant to generate highly predictive hypotheses for clinical decision-making with respect to any set of criteria.

Our long-term goal is to help clinicians review past clinical decisions with the insights from iTransplant and to improve future decisions of clinicians and suggestions from decision support tools. However, the analyses and experiments in our paper are mainly used to illustrate the potential usage of our method as a tool of understanding human decisions.

**Interpretation of the normalized policy signature**  As illustrated in Fig. 3 in our main text, the policy signatures identified by iTransplant indicate the relative importance (weights) of different

criteria in explaining the observed decisions. These weights can be used to discover the criteria or features that are correlated with clinicians' decisions in the observational dataset and can be interpreted as a set of potential drivers of human decisions. However, such correlations identified by our method should not be interpreted as causal, and validation by domain experts is always necessary for practical applications of inverse decision-making approaches, including our method.

**Interpretation of patient clusters**   The clustering of patients in the latent space learned by iTransplant is another important result of our method. By adopting the consistency loss in equation (2), similar latent representations of patients are encouraged to yield similar decision policies over the criteria space. Thus, clustering in the latent space can help us to group similar patients and explore correlations between patient features and certain decision drivers, which can provide insights into the observed decisions. However, we would like to emphasize that the insights discovered by our method should not be interpreted as causal, and careful validation by domain experts is always necessary.

## A.9   Potential negative social impacts

The proposed method iTransplant attempts to understand clinical decisions and does not directly recommend actions to clinicians. However, inaccuracies in the interpretations of iTransplant (or any inverse decision-making method) without proper audit could lead to negative consequences if they lead to changes in decision-making that adversely impacted outcomes, which should be taken into account in any practical use of such systems. For instance, misinterpreting the insights from our method as causal relationships between criteria and decision policies could lead to negative impacts on transplant decisions. Validation and additional analysis by domain experts is needed for practical application of our method and inverse decision-making methods more broadly.

Secondly, when patient or donor organ features related to (potentially) sensitive characteristics (e.g. gender or ethnicity) are included in the criteria space, analyses of biases learned by iTransplant are necessary in practice to avoid potential consequences on equality, fairness, etc. In addition, as a limitation of many inverse decision-making approaches, it is likely that some decisions were made based on factors that are not included in the observational dataset, which may bias the insights discovered by our method, while misspecification of the criteria space may also lead to incorrect interpretations.