# OpenReview forum: "Closing the loop in medical decision support by understanding clinical decision-making: A case study on organ transplantation"
_NeurIPS.cc/2021/Conference — NeurIPS 2021 Poster_

### Official Review · Reviewer_oN9u · 2021-07-02

**Rating:** 7
**Confidence:** 4

**Summary:**

In this paper, the authors propose a policy learning strategy that is patient-driven and interpretable. They focus on the case of organ transplant, not predicting matching but rather whether an organ will be accepted by a clinician for their patient. The authors show that their method perform on par with other approaches, with the added benefit that the decision space is interpretable and can be investigated.

**Limitations And Societal Impact:**

The authors provide a brief conclusion in the main text, and a short paragraph on limitations in the supplementary. I believe that the work is making a few assumptions and I would encourage the authors to explicitely discuss these in the main text. I have identified the following limitations (in addition to previous comments):
- The authors do not comment on a validation with clinicians as to whether the proposed approach would be useful for them to understand the decision making process.
- There is no validation that the proposed interpretability is useful. While SHAP and LIME are shown to provide results that are not consistent (which is not surprising given that these methods are not robust to perturbations), it is unclear whether clinicians would benefit from interpretability (in the sense that it helps them in achieving their goal), or that this technique in particular is answering their needs in terms of interpretability.
- Is the method able to handle missing data? How about other methods?
- Can the authors comment on some of the assumptions made, including linearity in the criteria space?
- The clustering and the investigation per center are interesting and convincing. However, there is little validation of the results, which are susceptible to confirmation bias. Can the authors comment or add as a limitation?

**Main Review:**

Originality
---------------
This is an interesting combination of different ML components to provide personalized policies. Technically, those components are not novel, but I find their combination interesting. The overall idea is quite similar (in philosophy) to that of Koh et al., 2020 (https://arxiv.org/abs/2007.04612). This parallel could be made explicit in the text.

Quality
----------
The experiments are mostly described in the supplementary materials. I have a few questions regarding some of the data and modeling choices:
- "For the experiments in this paper, we select rejected organ offers of twelve cases, covering 87.7% of all rejected offers". It feels like the data is selected based on parameters that are well represented in the criteria space. While I understand the reasons for not including all offers, this makes me question the broader applicability of the method. Can the authors comment on a potential circularity here, that might reduce the application to other settings or new patients?
- In the same way, the features for donors and patients are selected based on the criteria.
- The criteria space was selected based on domain knowledge. Do you mind expanding on who was consulted to provide those criteria? Was this set defined a priori and then used, or were multiple sets tried? If the latter, was there an effect on performance of the selected set?
- In the selection of hyper-parameters, the authors mention that they impose models to have similar learning capacities. This seems like an unnecessary constraint, especially given the diversity of approaches considered. Have the authors tried to optimize each technique, and maybe then compared the required numbers of parameters or CPU/GPU resources? While I understand the reasoning, a more fair comparison for overall usability would optimize each technique "independently" of the others, and then other parameters (such as computational expenses) can be taken into account in the comparison.
- Can the authors comment on the lower performance for center 6820?
- Supplementary Table 2 includes interesting results, showing that LASSO can perform similarly to the selected method when using (X,O) as input. Can the authors comment on their choice of including results in the main vs supplementary table?


Clarity
---------
The paper is overall well written. I would encourage the authors to revise some of the phrasings, that I found a bit convoluted. In general, simpler sentences would probably help the reader.
I also found that many of the experimental details were missing in the main text. I understand that these are part of the Supplementary Materials, but splitting the main text from the supplemental is a critical part of the writing, that I believe could be improved here (e.g. remove or make a bit more compact some of the intro and related works, bring in data descriptors, split and training/testing strategies at the high-level).
Another main source of confusion for me was that the space of criteria was not directly described as manually crafted based on domain knowledge. This seems to be critical for the interpretability of the method: a limited set of criteria, that are human-understandable. This information should be available early in the text, and discussed for its limitations.

Significance
-----------------
The authors mention that their work could help understand clinical decisions, and understand variations across centers. This seems promising but I was wondering whether those were desired by the clinical community? I also missed examples of other problems that could benefit from the method.

**Time Spent Reviewing:**

3

---

> ### Author Response · Authors · 2021-08-09
> **Response to Reviewer oN9u [Part 4/4]**
>
> [ References ]
>
> [1] M. R. Cooperberg, J. M.  Broering, and P. R. Carroll. Time trends and local variation in primary treatment of localized prostate cancer. Journal of clinical oncology : official journal of the American Society of Clinical Oncology, 28(7):1117–1123, 2010.
> [2] S. Hawley, T. Hofer, N. Janz, A. Fagerlin, K. Schwartz, L. Liu, and S. Katz. Correlates of Between-Surgeon Variation in Breast Cancer Treatments. Medical Care, 44(7):609-616, 2006.
> [3] C. W. Seymour, T. J. Iwashyna, W. J. Ehlenbach, H. Wunsch, and C. R. Cooke. Hospital-Level Variation in the Use of Intensive Care. Health Serv Res, 47:2060-2080, 2012.
> [4] F. Atsma, G. Elwyn, and G. Westert. Understanding unwarranted variation in clinical practice: a focus on network effects, reflective medicine and learning health systems, International Journal for Quality in Health Care, 32(4):271–274, 2020.
> [5] A. Rana, R. R. Sigireddi, K. J. Halazun, A. Kothare, M.-F. Wu, H. Liu, M. L. Kueht, J. M. Vierling, N. L. Sussman, A. L. Mindikoglu, et al. Predicting liver allograft discard: the discard risk index. Transplantation, 102(9):1520–1529, 2018.
> [6] L. H. Rosenberger, J. R. Gillen, T. Hranjec, J. B. Stokes, K. L. Brayman, S. C. Kumer, S. C., T. M. Schmitt, and R. G. Sawyer. Donor risk index predicts graft failure reliably but not post-transplant infections. Surgical infections, 15(2):94–98, 2014.
> [7] F. Atsma, G. Elwyn, and G. Westert. Understanding unwarranted variation in clinical practice: a focus on network effects, reflective medicine and learning health systems. International Journal for Quality in Health Care, 32(4):271–274, 2020.
> [8] D. S. Goldberg, B. French, J. D. Lewis, F. I. Scott, R. Mamtani, R. Gilroy, S. D. Halpern, and P. L. Abt. Liver transplant center variability in accepting organ offers and its impact on patient survival. Journal of Hepatology, 64(4):843–851, 2016.
> [9] J. E. Wennberg. Unwarranted variations in healthcare delivery: implications for academic medical centres. BMJ, 325(7370):961–964, 2002.
> [10] G. Thabut, J. D. Christie, W. K. Kremers, M. Fournier, and S. D. Halpern. Survival differences following lung transplantation among US transplant centers. JAMA. 304(1):53–60, 2010.
> [11] J. Godown, M. McKane, K. A. Wujcik, B. A. Mettler, and D. A. Dodd. 2017. Regional variation in the use of 1A status exceptions for pediatric heart transplant candidates: is this equitable?. Pediatric Transplantation, 21(1):e12784, 2017.
> [12] P. Gago, M. F. Santos, A. Silva, P. Cortez, J. Neves, and L. Gomes. INTCare: a knowledge discovery based intelligent decision support system for intensive care medicine. Journal of decision systems, 14(3):241-259, 2005.
> [13] J. Berrevoets, J. Jordon, I. Bica, M. van der Schaar, et al. Organite: Optimal transplant donor organ offering using an individual treatment effect. Advances in Neural Information Processing Systems, 33, 2020.
> [14] A. Flores, S. K. Asrani. The donor risk index: A decade of experience. Liver Transplantation, 23(9):1216-1225, 2017.
> [15] M. C. McCarthy, R. D. A. Lourenco, L. J. McMillan, E. Meshcheriakova, A. Cao, and L. Gillam. Finding out what matters in decision-making related to genomics and personalized medicine in pediatric oncology: developing attributes to include in a discrete choice experiment. The Patient-Patient-Centered Outcomes Research, 1-15, 2020.
> [16] V. J. Lozanovski, B. Döhler, K. H. Weiss, A. Mehrabi, and C. Süsal. The differential influence of cold ischemia time on outcome after liver transplantation for different indications—who is at risk? a collaborative transplant study report. Frontiers in Immunology, 11:892, 2020.
> [17] B. D. Ziebart, A. L. Maas, J. A. Bagnell, and A. K. Dey. Maximum entropy inverse reinforcement learning. In Aaai, volume 8, pages 1433–1438. Chicago, IL, USA, 2008.
> [18] D. Ramachandran and E. Amir. Bayesian inverse reinforcement learning. In IJCAI, volume 7, pages 2586–2591, 2007.

---

> ### Author Response · Authors · 2021-08-09
> **Response to Reviewer oN9u [Part 3/4]**
>
> [ 5. Limitations and societal impact ]
>
> Thank you very much for your insightful comments on the limitations. We have added further discussions on the assumptions of our work in the revision. The responses to your questions are given as follows.
>
> (5.1) "Validation of usefulness":
> We would like to first provide a clear statement on the interpretability proposed in our work. The interpretability of our method is inherited from the interpretability of the logistic regression model adopted in Eq. (1). Further, our method is able to provide personalized insights on human decisions without loss of such interpretability.
>
> In addition, we note that from a clinician’s perspective the true validation of usefulness would only come from a randomized trial, which is clearly beyond the scope of this paper. Given this, we address your comments on the validation of the usefulness of our method from two aspects as follows.
>
> (5.2.1) Usefulness for the clinical community:
> Logistic regression models are widely adopted in research outputs from the clinical community to analyse and interpret drivers of clinical decisions, including examples in the transplantation setting (e.g., [5, 6]). These logit model-based risk assessment tools are valued by the clinical community as tools to better understand risks associated with specific organ offers [14]. Since the proposed interpretability in our paper comes directly in the form of logit models, the interpretability of our method shares similar usefulness to these tools for the clinical community. In addition, the understanding of personalized decision drivers is a long-term focus of the clinical community [15, 16], which is in line with the proposed personalized interpretability of our method.
>
> (5.2.2) Usefulness for real clinicians:
> In addition, we have conducted a short survey with six clinicians in an attempt to further validate the usefulness of our method. The survey was conducted blindly, i.e., the clinicians were given no context as pertains to our proposed method. We asked clinicians several questions regarding the interpretability and usefulness of several different types of interpretation. Notably, all six clinicians deemed individualized logistic regression models more informative and practical than a general logistic regression model. In addition, five out of the six surveyed clinicians consider logistic regression models more interpretable and helpful than post-hoc interpretations of black-box models.
>
> Despite the relatively small sample size, we believe that these results provide evidence of the importance of interpretability in our proposed approach and the usefulness of our method compared to alternate forms of interpretability, including basic logistic regression models.
>
> (5.3) Missing data:
> We agree that missing data is an important issue for practical applications of machine learning. That said, our proposed method tackles a problem orthogonal to that of missing data. To be clear, neither our method nor any of the included benchmarks have any "built-in" capabilities designed for missing data. However, in practice our method (and benchmarks) is compatible with any existing data imputation algorithms. Finally, allow us to emphasize that in our experiments, the impact from data missingness is negligible: for the data of patient and donor organ features utilized in our experiments, the average data missing rate over the ten considered transplant centers is 3.42% with the maximum missing rate of 6.09% for center 23808.
>
> (5.4) Comments on assumptions made:
> We would like to clarify that all our assumptions are already discussed in the manuscript. For added clarity, we expand on these here. In addition, this additional discussion will be included in an auxiliary section in the Supplementary Materials.
>
> (5.4.1) Linearity in the criteria space:
> The assumption of linearity in the criteria space (or the reward structure) is common in inverse decision-making literature that aims to identify interpretable decision policies from the observable decision history of an agent, see, e.g., [17, 18]. We take the same assumption of linearity to obtain human comprehensible decision policies. However, the validity of this assumption strongly depends on the selection of feature maps (criteria) in the criteria space. Misspecification of the criteria space may lead to misleading interpretations and poor performance of inverse decision-making methods. Thus, careful validation by domain experts is necessary for practical applications.
>
> (5.4.2) Greedy decision policy:
> The OPTN database describes donor organ-recipient pairs, where the organ was offered according to national guidelines to an individual recipient, and accepted or declined by a clinician. The decision to accept or decline was not informed by knowledge of the whole waitlist of potential recipients nor the future availability of donor organs. In line with this, we take the assumption that the decision sequences of clinicians are greedy and not purposeful (accounting for outcomes of future decisions). The assumption of greedy decision policy is necessary for modeling the organ offer acceptance as a one-step decision-making problem and was deemed reasonable in our discussions with clinicians.
>
> (5.4.3) Personalized reward function:
> We assume that clinicians' decisions on organ offers are customized for patients from different cohorts. This is a reasonable assumption since the outcomes of clinicians’ decisions are highly related to the characteristics of their patients. Please refer to the investigative experiments in the main text for examples from the medical literature.
>
> (5.4.4) Logit model of decision policy:
> As a "soft" version of the $\\mathrm{argmax}$ operator, it is a common practice in the inverse decision-making literature to model the decision policy of the decision maker as a Boltzmann distribution (logit model in our case) (see, e.g., [17, 18]). The formulation in Eq. (1) ensures that the decision policy is differentiable.
>
> (5.4.5) No hidden confounders:
> It is assumed that drivers of observed decisions in the dataset are fully observable or can be inferred from observable features in the dataset. However, as a limitation of many inverse decision-making approaches, it is likely that some decisions were made based on factors that were not included in the observational dataset, which may bias the insights discovered by our method and lead to lower prediction performance. This limitation is discussed in L340-343 of the main text.
>
> (5.5) Confirmation bias:
> We agree that any endeavor aimed at "pattern discovery" will run the risk of confirmation bias. In most realistic use cases, there will be no absolute "ground truth" available for validation, and our inverse decision-making approach is not immune to this limitation. Importantly, however, we emphasize that we are *not* in the business of performing statistical inference or testing (whence the formal quantification of "statistical significance" would require rigid procedures to ensure validity). To the contrary, we are engaged in the orthogonal mission of generating interpretable hypotheses on the underlying drivers of human decisions---in particular, drivers that are highly predictive of observed decisions. The purpose of the investigative experiments is to illustrate that our method is capable of proposing *hypothetical* decision policies that could be matched with real clinical findings or guidelines, which makes our method useful for clinicians to review and improve their decisions. In the meantime, for practical applications of our method, validation and additional analysis by domain experts is necessary to validate the discovered insights.

---

> > ### Comment · Reviewer_oN9u · 2021-08-12
> > **Response to the authors**
> >
> > I would like to thank the authors for their thorough responses and for the extra work.
> >
> > I appreciate the extra experiments with expanded hyper-parameter space, as well as the contrast across feature sets.
> >
> > Most of my questions have been answered and the inclusion of further discussion and limitations satisfy me.

---

> ### Author Response · Authors · 2021-08-09
> **Response to Reviewer oN9u [Part 2/4]**
>
> [ 2. Quality (Continued) ]
>
> (2.6) Choice of results:
> We agree with the reviewer that the presentation of our experiment results could be better explained in our manuscript. To be clear, we can provide the results with both input spaces consistently. As for our choice of benchmark results, Table 2 in the main text is mainly used to evaluate the prediction performance of our method with a set of inherently interpretable baselines under the same input space ($\\mathcal{X} \\times \\mathcal{M}$). The blackbox models of BC and INVASE were included in Table 2 to provide the upper performance bounds that could be achieved with the information available in our dataset. Due to the black-box nature of BC and to provide strong benchmarks, the results of BC and INVASE originally included in Table 2 of the main text used a larger input space ($\\mathcal{X} \\times \\mathcal{O}$). To ensure that this choice did not impact the conclusions of model validation, we conducted the additional experiments reported in Table 2 of the Supplementary Material. As can be seen, the performance of INVASE, BC, and iTransplant are all virtually unchanged. This also serves as validation of the choice of criteria space, which contains substantially fewer features.
>
> We appreciate and apologize that this is not clear in our manuscript and will revise our results tables accordingly to ensure there is no confusion in the final version of our manuscript. As such, all results in the updated Table 2 will be with respect to this same input space $\\mathcal{X} \\times \\mathcal{M}$ (see our response (2.4) for these results), while we will also include a full set of results with $\\mathcal{X} \\times \\mathcal{O}$ as input space. Please note that none of our conclusions are affected by this change. Finally, although the performance of LASSO improves with the additional donor organ features as input, it is still significantly worse than the best performing methods. Please refer to (2.7) for details.
>
> (2.7) Improvements in the performance of LASSO:
> We thank the reviewer for highlighting this interesting result. First, although the performance of LASSO improves with the additional donor organ features as input, it is still significantly worse than the best performing methods (in particular, see the performance as measured by AUC-PRC which takes into account the imbalanced nature of the data). Second, we have investigated the source of performance gains of LASSO by comparing the coefficients learned by LASSO when $\\mathcal{X} \\times \\mathcal{O}$ and $\\mathcal{X} \\times \\mathcal{M}$ are used as model input, respectively. The result shows that many important donor organ features (with the absolute value of coefficients greater than 0.01) discovered by LASSO are already included in the criteria space $\\mathcal{M}$. However, there are several additional features that appear to be informative (bilirubin and creatinine concentrations of donor, blood type of donor, cause of death of deceased donor and death mechanisms of donor) not included in the criteria space $\\mathcal{M}$ used in our illustrative experiments, which explains the improvement in the performance of LASSO when $\\mathcal{X} \\times \\mathcal{O}$ is used as input.
>
> We have conducted a new experiment to evaluate the benefit of including these additional features into the criteria space $\\mathcal{M}$ for iTransplant. For the hyperparameters used in this experiment, please refer to (2.4) and the Supplementary Material. A comparison of the experimental results for LASSO and iTransplant are given in the table below.
>
> | Method       | Input space                                     | AUC-ROC    | AUC-PRC    | NLL                |
> |:----------------|:----------------------------------------------|:-----------------|:-----------------|:-------------------|
> | LASSO       | $\\mathcal{X}\\times\\mathcal{M}$ | 0.792±0.054 | 0.332±0.060 | -0.568±0.043 |
> | LASSO       | $\\mathcal{X}\\times\\mathcal{O}$ | 0.877±0.051 | 0.444±0.064 | -0.477±0.048 |
> | iTransplant | $\\mathcal{X}\\times\\mathcal{M}$ | 0.896±0.044 | 0.501±0.060 | -0.394±0.072 |
> | iTransplant | $\\mathcal{X}\\times\\mathcal{O}$ | 0.895±0.050 | 0.505±0.063 | -0.390±0.078 |
>
> The performance of iTransplant slightly improves when all donor organ features are present in the criteria space, and significantly outperforms LASSO with either input space. Since the performance of iTransplant with both input spaces is comparable, we believe the criteria space used in our paper is sufficient for the purpose of our investigative experiments. However, for real-world applications, the design of the criteria space needs to be carefully considered.
>
> [ 3. Clarity ]
>
> (3.1) “Convoluted sentences”:
> Thank you for your suggestions. We have simplified the phrasings in our revised manuscript to improve the overall clarity.
>
> (3.2) "Many of the experimental details were missing in the main text":
> We agree that more experimental details should be included in the main text for better readability of our work. We have improved the clarity of our experimental results by including additional descriptions and explanations in our revision.
>
> (3.3) "The space of criteria was not directly described as manually crafted based on domain knowledge":
> We thank the reviewer for their comment. We will ensure the description of the criteria space and the selection of criteria in our revised paper is clearer. We would like to clarify that we did explicitly state that the criteria space was manually crafted based on domain knowledge at the start of Section 5 (L221-223). Of note, our method works with larger criteria spaces (e.g., the whole organ space without extra domain knowledge, see (2.7)) and could be used by clinicians as a generator of highly predictive hypotheses on clinical decisions with any set of criteria to gain more insights from the observational data. Please refer to (5.5) for more details. We have included additional discussion in our manuscript to make this point clearer as well.
>
> [ 4. Significance ]
>
> (4.1) "Whether those were desired by the clinical community":
> In the introduction of our manuscript, we extensively describe a number of challenges faced by the clinical community and the importance of understanding variations in clinical decision-making. This is particularly true in transplantation but is well studied across medicine (e.g., cancer [1, 2], intensive care [3]). In particular, a recent paper [4] explicitly states that “a better understanding of causes of variation should be gained”. This is precisely the aim of our manuscript, and understanding clinical variations in transplantation is a focus of many transplant jurisdictions. We appreciate the reviewer’s comments on this topic and have provided further discussion on the application areas of our work in our revised manuscript. We further elaborate our answer as follows.
>
> (4.1.1) Understand clinical decisions:
> Taking organ transplantation as an example, both the discard risk index [5] and donor risk index [6] were developed by the clinical community to better understand the outcome of an offer (the donor discard index) and the outcome of the transplant (donor risk index). While both differ from our aim of understanding the donor or recipient factors potentially associated with decisions on accepting or rejecting a donor, they serve as further evidence as to the potential clinical relevance of our work.
>
> (4.1.2) Understand variations across transplant centers:
> Similarly, there is a lot of relevant medical research on variations of practices and outcomes across transplant centers from the clinical community (see, e.g., [7, 8, 9, 10]). Some variations may depend on residual donor and/or recipient factor confounding, and are not adequately accounted for in risk adjusting. Thus, the understanding of variation in clinical decision-making between centers could help clinicians to improve their future decisions and potentially reduce the variability in outcomes for their patients. In addition, studies on the cross-center variations could help to inform the organ allocation policies and improve the fairness of the allocation system [11].
>
> (4.2) "Other problems that could benefit from the method":
> It is evident that many clinical scenarios are impacted by clinical decision-making, ranging from decisions on when or if to start antibiotic therapy to when and if to utilize an intensive care bed. Many such scenarios exhibit widespread variations in outcomes with different clinical practices. The insights on clinical decision-making provided by our method could be beneficial to study such variations. For instance, the hypothetical decision policies generated by our method can be utilized in automated knowledge discovery for clinical decision-making [12]. With careful application and validation by domain experts, the results from iTransplant could also be used to inform the development of clinical decision support tools [13]. In addition, the ability of our method to detect policy variations would be helpful for clinical studies focusing on the impact of variation in clinical decision-making [1, 2, 3]. We have also included additional discussion on the broader impacts of our method in the revision of our manuscript.

---

> ### Author Response · Authors · 2021-08-09
> **Response to Reviewer oN9u [Part 1/4]**
>
> Thank you for the helpful comments and suggestions. We give responses to each of them in turn.
>
> [ 1. Originality ]
>
> (1.1) "The overall idea is quite similar (in philosophy) to that of Koh et al., 2020 (https://arxiv.org/abs/2007.04612). This parallel could be made explicit in the text.":
> Thank you for bringing this reference to our attention. We agree that the idea of mapping the input features into a concept space and performing prediction tasks using the concept space is to some extent similar to our proposed method in philosophy. We have included this paper in the reference of our revised manuscript. However, we would like to point out that the target of our method differs significantly from Koh et al. They aim to predict a set of human-specified concepts as an intermediate step, with each concept having a fixed (albeit learned) importance on the final prediction; contrastingly, we aim to predict a policy (equivalently a set of weights) over a set of clinician-specified criteria, but not the values of such criteria. Additionally, their method focuses on prediction tasks such as image classification and X-ray image grading, while our proposed method aims at discovering the underlying drivers of clinicians' decisions in organ transplantation. The distinct targets lead to significant differences in the problem formulation, methodology and analysis in our paper and that of Koh et al.
>
> [ 2. Quality ]
>
> (2.1) Data selection:
> We thank the reviewer for the comments. As described in the Supplementary Material (L18-21), in a minority of cases, the reasons for organ offer rejection are either administrative in nature (e.g., “surgeon unavailable” and “heavy workload”) or do not directly relate to possible patient or organ factors available in our datasets (e.g., “patient’s condition improved, transplant not needed”). This is the reason why some rejected offers are excluded from our experiments. The decision regarding which organ offers to include were made in consultation with clinicians, including one who is a co-author of our paper. Regarding your question on the applicability, our method is applicable to the settings where the underlying decision drivers are observed as feature variables in the dataset or can be inferred from other variables available in the observational dataset. As such, we believe our method is broadly applicable and do not believe there is circularity.
>
> (2.2) Selection of features and criteria:
> The features and criteria were selected based on the features available in the considered organ transplantation database (OPTN) in consultation with clinicians working in organ transplantation, including one who is a co-author of the paper. Their help and assistance will be explicitly acknowledged in the final version of our paper.
>
> (2.3) "Was the set of criteria defined a priori and then used":
> Yes. As described in our problem formulation in Section 2 (L101-106), the set of criteria is defined a priori.
>
> (2.4) Hyperparameter selection:
> Thank you very much for your suggestion on hyperparameter selection. Note that the hyperparameters of logistic regression, decision tree, random forests, etc. are already optimized individually for each method. The hyperparameters of these methods are kept unchanged in our revision. We followed your advice and reconducted hyperparameter selection for iTransplant, behavioral cloning (BC), and INVASE with the number of layers $l_n$ for each module and the number of hidden units $h_n$ in each layer as additional hyperparameters. In addition, the hyperparameter selection of locally weighted regression (LOESS) is reconducted with a new range of parameters. The hyperparameter selection process remains unchanged as previously, and the ranges of hyperparameters considered for each method are shown as follows.
> - BC: $l_n \\in \\{2,4,6,8\\}$, $h_n \\in \\{10,20,30,40,50\\}$
> - INVASE: $l_n \\in \\{2,4,6\\}$, $h_n \\in \\{20,30\\}$, $\\lambda \\in \\{0.01, 0.1,0.5\\}$.
> - iTransplant: $l_n \\in \\{1,2,3\\}$, $h_n \\in \\{30,40,50\\}$, $\\lambda \\in \\{0.01,0.05\\}$, $K \\in \\{10,20\\}$, $k \\in \\{4,8\\}$.
> - LOESS: $\\sigma \\in \\{0.01,0.1,1,10\\}$
>
> The hyperparameter selection outcomes are given as follows.
> - BC: $l_n=2$, $h_n=50$.
> - INVASE: $l_n=6$, $h_n=30$, $\\lambda=0.5$.
> - iTransplant: $l_n=2$, $h_n=50$, $\\lambda=0.01$, $K=10$, $k=4$.
> - LOESS: $\\sigma=1$
>
> The updated benchmark in Table 2 of the main text is given below. We would like to emphasize that the hyperparameters are optimized for negative log-likelihood (NLL), and the input space for all methods in the table below and in Table 2 in our revised manuscript are the same ($\\mathcal{X} \\times \\mathcal{M}$).
>
> | Method                                                           | AUC-ROC     | AUC-PRC    | NLL                |
> |:-----------------------------------------------------------|:------------------|:------------------|:------------------|
> | LOGISTIC REGRESSION                             | 0.794±0.054  | 0.341±0.061 | -0.538±0.051 |
> | PER-CLUSTER LOGISTIC REGRESSION   | 0.817±0.057 | 0.358±0.076 | -0.512±0.076 |
> | LOCALLY WEIGHTED REGRESSION          | 0.866±0.044 | 0.426±0.065 | -0.256±0.089 |
> | DECISION TREE                                           | 0.775±0.057 | 0.274±0.069 | -0.552±0.060 |
> | PER-CLUSTER DECISION TREE                 | 0.776±0.041 | 0.285±0.072 | -0.586±0.069 |
> | RANDOM FOREST                                        | 0.852±0.063 | 0.421±0.105 | -0.271±0.092 |
> | LASSO                                                            | 0.792±0.054 | 0.332±0.060 | -0.568±0.043 |
> | INVASE                                                           | 0.882±0.041 | 0.474±0.094 | -0.439±0.141 |
> | BEHAVIORAL CLONING                                | 0.904±0.040 | 0.523±0.062 | -0.370±0.067 |
> | iTransplant                                                      | 0.896±0.044 | 0.501±0.060 | -0.394±0.072 |
>
> As can be seen in the above table, our conclusions remain unchanged. iTransplant significantly outperforms all baseline methods, and has similar performance to BC, while maintaining interpretability of policies identified for each patient.
>
> We have updated the contents relevant to hyperparameter selection in the revised version of our paper accordingly.
>
> (2.5) "Lower performance for center 6820":
> Following your suggestions regarding hyperparameter selection, we have re-evaluated the per center performance of our method. With updated hyperparameters, the performance of iTransplant is similar to BC on all of the ten considered transplant centers (performance gaps: AUC-ROC 0.008(mean), 0.017(max); AUC-PRC 0.022(mean), 0.047(max)), including center 6820, for which the performance was lower in our original benchmark results. The updated performance comparison between our method and BC on center 6820 is as follows:
>
> | Method                | AUC-ROC    | AUC-PRC    | NLL                |
> |:-----------------------|:------------------|:-----------------|:-------------------|
> | BC                       | 0.892±0.003 | 0.502±0.015 | -0.399±0.015 |
> | iTransplant          | 0.884±0.006 | 0.474±0.018 | -0.442±0.021 |

---

### Official Review · Reviewer_Q9SD · 2021-07-05

**Rating:** 7
**Confidence:** 4

**Summary:**

This paper proposes a ML procedure (or framework) for understanding donor organ acceptance decisions for patients on an organ waiting list. Their framework uses NNs for both learning a latent space (encoder/decoder), and for identifying policy criteria for the patient decision function. They include a variety of constraints (consistency, partial monotonicity, model complexity) that are salient to a real organ allocation setting. They lightly discuss their results on real data from UNOS.

**Ethical Concerns:**

See "Limitations and societal impact".

I don't want to flag this paper for ethics review. But I would certainly like the authors to address the questions I raised.


**Limitations And Societal Impact:**

I'm a bit concerned about the interpretation and use of the methods proposed in this paper, and I think more discussion is needed.
- What is the intended use of this model?
- What is the interpretation of your results? (Especially the normalized policy signature in Fig. 3 and the patient clusters.)
- Is this analysis meant to influence future decisions? And if so, how does this differ from state-of-the-art decision tools for organ allocation?
    - If so, what are the risks for using this method?

These are issues because of the sensitive and applied nature of this paper, where the prediction task at hand is literally tied to human life and death.

**Main Review:**

This paper is very well written and mostly easy to follow. Their main contributions are relatively novel, and they do improve on prior work in some meaningful ways. In particular: Their model is useful for illustrating how individual-level (and hospital-level) organ accept/reject decisions are made, and which latent features are most salient to these decisions. This extends prior work with "black-box" models, which provide no intuition as to the decision making process. This interpretability comes at very little cost in accuracy, since their model performs nearly as well as the best "black box" model they compare against.


Some general comments & questions follow:

- The latter part of the pipeline is somewhat unclear: what role does the MoE module play, and why use an MoE rather than a single network? (It seems that this module learns the transform \mathcal{T} and weight vector(?) \rho, however this is not clear.)

- In general, there are several design decisions that seem arbitrary, yet are very important to the model design. I think it would substantially improve the readability and impact of this paper if you clarified these design choices when introducing the model. Some of these choices include:

    - Why choose the Boltzman distribution / logit model in (1), as opposed to some other model? (This model is similar or identical to a multinomial logit, which would be natural appropriate if a patient needs to choose between multiple organs. But in your case, a patient is choosing whether or not to accept a single organ, or wait for another option. So what is the intuition behind this model?)

    - Why use the MoE module as opposed to something else?

    - Why is consistency an important criteria? (Eq. (2)). There are many reasons my preferences for donor organs would not be smooth -- for example, I may be likely to reject an organ from a donor with a particular antibody that I am incompatible with, even though this donor may be a good fit in all other aspects. (So this organ is close to a desirable organ in feature space, but very far from the desirable organ in decision space.)

To be clear, I'm not saying that these choices are bad. I just think you need to justify them.

- Some comments on the application & related literature. Overall, since this is mainly an applications paper, I think there needs to be more discussion on related application-focused literature as well as more explanation of the experiments.

    - There is a large body of literature on this topic, and I suggest you engage with it more in your paper. For example, this paper (https://aasldpubs.onlinelibrary.wiley.com/doi/full/10.1002/lt.24113) has a very similar goal to your paper, with a more applied focus. They discuss two main criteria of interest to organ patients: graft failure and disease transmission. It is likely that these factors also play a major role in the decisions you are aiming to model in your paper. If you do not use this domain knowledge, you're just making the task of predicting donor behavior harder. Even if you do not use this domain knowledge, you should certainly cite it.

    - The organ accept/reject decisions are strongly dependent on both the waitlist prioritization (who has the highest priority to receive an organ) and the compatibility criteria (when is an organ compatible with a patient). At very least, these policies will make certain features in your dataset very dependent. For example, certain patient types will never be offered certain types of donor organ. Please discuss how these policies impact your analysis.



**Time Spent Reviewing:**

2

---

> ### Author Response · Authors · 2021-08-09
> **Response to Reviewer Q9SD [Part 2/2]**
>
> [ 4. Limitations and societal impact ]
>
> Thank you very much for your comments. We have included detailed discussion on the intended use of our method and the associated potential negative impacts in our revised manuscript.
>
> (4.1) "What is the intended use of this model":
> As mentioned in our main manuscript and the Supplementary Material, our model is proposed as a tool to help clinicians to review and identify potential drivers of their clinical decisions. As described in the introduction to our manuscript, clinical decision-making in organ transplantation is poorly understood with a high proportion of organ offers declined and substantial variation in clinical practice [6]. The intended use of a tool such as iTransplant would be to stand beside the clinician decision-makers so that they can reflect upon and better understand the drivers for their decision. These issues are not unique to liver transplantation, and variation in practice has been studied across medicine [7,8], including cancer [9, 10] and intensive care [11].
>
> As illustrated in our experiments, our method can be used to study and gain greater understanding of these clinical phenomena, and the insights obtained via our method could ultimately be used to improve future decisions. It is worth noting that the criteria space in our method is not limited to the one discussed in our paper. In fact, clinicians could use iTransplant to generate highly predictive hypotheses for clinical decision-making with respect to any set of criteria. We have added additional discussion in our manuscript to clarify this point.
>
> (4.2) Interpretation of the normalized policy signature:
> As illustrated in Fig. 3 in our main text, the policy signatures identified by iTransplant indicate the relative importance (weights) of different criteria in explaining the observed decisions. These weights can be used to discover the criteria or features that are correlated with clinicians' decisions in the observational dataset and can be interpreted as a set of potential drivers of human decisions. However, such correlations identified by our method should not be interpreted as causal, and validation by domain experts is always necessary for practical applications of inverse decision-making approaches, including our method.
>
> (4.3) Interpretation of patient clusters:
> The clustering of patients in the latent space learned by iTransplant is another important result of our method. By adopting the consistency loss in Eq. (2), similar latent representations of patients are encouraged to yield similar decision policies over the criteria space. Thus, clustering in the latent space can help us to group similar patients and explore correlations between patient features and certain decision drivers, which can provide insights into the observed decisions. However, similar to the discussion in (4.2), we would like to emphasize that the insights discovered by our method should not be interpreted as causal, and careful validation by domain experts is always necessary.
>
> (4.4) "Is this analysis meant to influence future decisions":
> Our long-term goal is to help clinicians review past clinical decisions with the insights from iTransplant and to improve the future decisions of clinicians and suggestions from decision support tools. However, the analyses and experiments in our paper are mainly used to illustrate the potential usage of our method as a tool of understanding human decisions.
>
> (4.5) "How does this differ from state-of-the-art decision tools for organ allocation":
> Firstly, we would like to clarify that iTransplant is not intended as a tool for better organ allocation, but was developed to better understand clinicians’ decisions when offered an organ. Thus, we do not consider the organ allocation step in our work. Second, there are significant distinctions between the aim of our method and state-of-the-art decision tools for organ allocation. Our method focuses on the inverse problem of decision-making (understanding of decision drivers) while existing decision tools consider the forward problem, i.e., optimizing suggested decisions from their proposed algorithms for higher potential outcomes (see, e.g., [12, 13]).
>
> (4.6) "What are the risks for using this method":
> The risks from using our method are largely shared by any inverse decision-making system. Firstly, as mentioned in (4.2) and (4.3), misinterpreting the insights from our method as causal relationships between criteria and decision policies could lead to negative impacts on transplant decisions. Validation and additional analysis by domain experts is needed for practical application of our method and inverse decision-making methods more broadly. Second, as a limitation of many inverse decision-making approaches, it is likely that some decisions were made based on factors that are not included in the observational dataset, which may bias the insights discovered by our method, while misspecification of the criteria space may also lead to incorrect interpretations (see L340-343 of the main text). We have included additional discussion regarding potential risks of the use of our method in the revised version of our paper.
>
> [ References ]
>
> [1] B. D. Ziebart, A. L. Maas, J. A. Bagnell, and A. K. Dey. Maximum entropy inverse reinforcement learning. In Aaai, volume 8, pages 1433–1438. Chicago, IL, USA, 2008.
> [2] D. Ramachandran and E. Amir. Bayesian inverse reinforcement learning. In IJCAI, volume 7, pages 2586–2591, 2007.
> [3] H. Yersiz, C. Lee, F. M. Kaldas, J. C. Hong, A. Rana, G. T. Schnickel, J. A. Wertheim, A. Zarrinpar, V. G. Agopian, J. Gornbein, et al. Assessment of hepatic steatosis by transplant surgeon and expert pathologist: a prospective, double-blind evaluation of 201 donor livers. Liver Transplantation, 19(4):437–449, 2013.
> [4] A. Rana, R. R. Sigireddi, K. J. Halazun, A. Kothare, M.-F. Wu, H. Liu, M. L. Kueht, J. M. Vierling, N. L. Sussman, A. L. Mindikoglu, et al. Predicting liver allograft discard: the discard risk index. Transplantation, 102(9):1520–1529, 2018.
> [5] M. L. Volk, N. Goodrich, J. C. Lai, C. Sonnenday, K. Shedden. Decision support for organ offers in liver transplantation. Liver transplantation, 21(6), 784-791, 2015.
> [6] D. S. Goldberg, B. French, J. D. Lewis, F. I. Scott, R. Mamtani, R. Gilroy, S. D. Halpern, and P. L. Abt. Liver transplant center variability in accepting organ offers and its impact on patient survival. Journal of Hepatology, 64(4):843–851, 2016.
> [7] J. E. Wennberg. Unwarranted variations in healthcare delivery: implications for academic medical centres. BMJ, 325(7370):961–964, 2002.
> [8] F. Atsma, G. Elwyn, and G. Westert. Understanding unwarranted variation in clinical practice: a focus on network effects, reflective medicine and learning health systems. International Journal for Quality in Health Care, 32(4):271–274, 2020.
> [9] M. R. Cooperberg, J. M. Broering, and P. R. Carroll. Time trends and local variation in primary treatment of localized prostate cancer. Journal of clinical oncology : official journal of the American Society of Clinical Oncology, 28(7):1117–1123, 2010.
> [10] S. Hawley, T. Hofer, N. Janz, A. Fagerlin, K. Schwartz, L. Liu, and S. Katz. Correlates of Between-Surgeon Variation in Breast Cancer Treatments. Medical Care, 44(7):609-616. 2006.
> [11] C.W. Seymour, T. J. Iwashyna, W. J. Ehlenbach, H. Wunsch, and C. R. Cooke. Hospital-Level Variation in the Use of Intensive Care. Health Serv Res, 47:2060-2080, 2012.
> [12] J. Yoon, A. Alaa, M. Cadeiras, and M. Van Der Schaar. Personalized donor-recipient matching for organ transplantation. In Proceedings of the AAAI Conference on Artificial Intelligence, volume 31, 2017.
> [13] J. Neuberger, A. Gimson, M. Davies, M. Akyol, J. O’Grady, A. Burroughs, M. Hudson, U. Blood, et al. Selection of patients for liver transplantation and allocation of donated livers in the UK. Gut, 57(2):252–257, 2008.

---

> ### Author Response · Authors · 2021-08-09
> **Response to Reviewer Q9SD [Part 1/2]**
>
> Thank you very much for your thoughtful comments. We provide a point-by-point response below.
>
> [ 1. Clarity of our paper ]
>
> (1.1) "The latter part of the pipeline is somewhat unclear":
> We thank the reviewer for the comments and suggestions on clarity. We are willing to amend Section 3 of our paper to improve the readability and clarify the model design choices of our proposed method. In addition, we have added more explanations of the experimental details to the main text to improve the overall clarity of our paper.
>
> [ 2. Model design choices ]
>
> We thank the reviewer for the comments on our model design, and the points discussed below have been included in Section 2 & 3 of our revised paper.
>
> (2.1) "Why choose the Boltzmann distribution / logit model in (1), as opposed to some other model":
> We agree that the decision policy in Eq. (1) could be any function that maps the criteria space to the binomial distribution. However, there are two reasons behind our choice of the Boltzmann distribution / logit model in Eq. (1):
>
> (2.1.1) The assumption of Boltzmann distribution is commonly used in inverse decision-making:
> As a "soft" version of the $\\mathrm{argmax}$ operator, it is common practice in the inverse decision-making literature to model decision policies of the decision maker as a Boltzmann distribution (see, e.g., [1, 2]). This formulation makes sure that the decision policy is differentiable.
>
> (2.1.2) The logit model is one of the simplest white-box models that is compatible with our assumptions:
> The ultimate goal of our method is to provide human comprehensible insights of decisions to clinicians. The logit model is one of the simplest linear models that satisfies the assumption of partial monotonicity (Assumption 2). More importantly, the logit model is widely adopted in medical research (see, e.g., [3, 4]), which makes it relatively easy to be understood and accepted by the clinical community.
>
> (2.2) "Why use the MoE module as opposed to something else":
> As illustrated in Figure 1 of the Supplementary Material, the gating network in the MoE layer tends to assign different sets of expert networks for patients from different patient clusters.
> This suggests that the MoE layer encourages the encoder network to group patients sharing the same decision policy in the latent space, which is a desired property of iTransplant.
>
> (2.3) "Why is consistency an important criterion":
> Thank you for your comments on consistency. We believe this is a very important topic and provide two justifications below.
>
> (2.3.1) Why consistency is important:
> Consistency is proposed as one of the key desiderata of practical inverse decision-making approaches in Section 2 of our paper. For the insights from inverse decision-making methods to be helpful for decision makers, the interpretations need to be consistent under moderate perturbations in input space. Practically, this means that if a feature has a positive weight, a moderate increase to the value of the feature should increase the probability of acceptance (and for a binary feature with positive weight, flipping the feature should reduce the probability of acceptance). Without this property, counterfactual analysis cannot be reliably performed, limiting the value of any interpretation or understanding gained from inverse decision-making. Further discussion on consistency can be found in Section 5.1 (L240-254) of our paper.
>
> (2.3.2) Non-smooth preferences:
> We completely agree with your point regarding non-smooth preferences for donor organs. We would like to clarify that the consistency loss defined in Eq. (2) enforces consistency between latent space representations and policies, rather than consistency between the input features and policies. Altering patient features could drastically change the latent representation, thus changing the policy significantly, which is the desired outcome in our model. In addition, we note that the logit model in Eq. (1) allows a single criterion in the criteria space to flip the decision and thus our modeling framework can accommodate the preferences described in your example scenario.
>
> [ 3. Review of application & related literature ]
>
> Thank you very much for your comments and suggestions on this. We have updated the related work section of our paper and added more discussion on application-focused literature. We have provided more explanations of the experiments in our revised paper as well.
>
> (3.1) "This paper (https://aasldpubs.onlinelibrary.wiley.com/doi/full/10.1002/lt.24113) has a very similar goal to your paper, with a more applied focus":
> Thank you for the reference of this related work, and we have included this paper in the revised version of our manuscript. However, we would like to emphasize that the focus of our paper differs from this work. In [5], the authors developed a decision support tool to predict whether an organ offer should be accepted by optimizing covariate-adjusted survival probabilities. In contrast, the goal of our method is to discover potential drivers behind the observed decisions of clinicians. In the meantime, we agree with the reviewer that the two main criteria of graft failure and disease transmission considered in [5] are highly relevant and indeed are (at least partially) captured by some of the criteria considered in our paper. To this point, the features and criteria used were selected in consultation with several clinicians working in organ transplantation, including one who is a co-author of the paper. Thus, we believe we have appropriately incorporated domain knowledge in the design of our illustrative experiments, as encouraged by the reviewer.
>
> (3.2) "How organ offering policies impact your analysis":
> We thank the reviewer for their excellent points and have added further discussion on this issue in the revised version of our manuscript. Currently, specific organs are offered to clinicians for a specific recipient, with that offer being predicated on nationally agreed guidelines. A clinician will accept or reject the offer based on their knowledge, experience and discussion with the recipient. Nevertheless, transplant centers have very variable acceptance rates of offered organs, with some consistently accepting a higher percentage than others. Hence, indeed, the organ offering policy determines which organ offers clinicians must make decisions on, which ultimately shapes the *observed* distribution of organ offers in the data. However, the goal of our method is precisely to understand the *observed* decisions of clinicians---i.e., given the organ allocation policy in the real transplantation system.
>
> To this end, we seek interpretable parameterizations of the drivers of human decisions that are predictive of these observed decisions. Of course, these parameterizations are only valid for the observed data (i.e., "in-distribution" prediction), and should *not* be used in scenarios where the organ offering policy is significantly different (i.e., "out-of-distribution" prediction). In this sense, this caveat is no different from any inverse decision-making or supervised learning setting---a change in domain will generally degrade the analysis.

---

### Official Review · Reviewer_PPKT · 2021-07-16

**Rating:** 7
**Confidence:** 3

**Summary:**

Develop a data-driven policy learning framework to infer clinical decision making processes around the applied question of organ transplant offers and acceptances (organ donation data). Use this approach to better understand the underlying decision processes with mroe interpretability than "black box" methods, but more personalization and accuracy than standard logistic regression methods.


**Ethical Concerns:**

No major issues. Research unpacks potentially high stakes transplant decisions that could inadvertently bias against underrepresented groups. Overall the purpose to gain better interepretability into black box physician decision making processes may be beneficial to expose likely existing biases in routine (but variable) practice.


**Limitations And Societal Impact:**

Hard to objectively compare the proposed method vs. existing methods.
Better understanding of decision drivers and risk assessment could significantly influence organ transplant practices. Misinterpreting correlations found through these approaches as causal could negatively impact transplant decisions without clear guidance on interpretation of findings.

**Main Review:**

Note that I already reviewed a prior version of this paper that was submitted to ICML. I overall rated the manuscript favorably then and will mostly reiterate my assessment here:

Good description of a high-stakes but variable clinical decision problem and a policy modeling framework to better understand sources of that variability.

Since the goal is not achieving a particular measurable outcome (e.g., prediction accuracy), not as clear how to judge whether the interpretability of suggestions here offers decision makers much more value than they could get from logistic regression models.

Argues that logistic regression is too general a model vs. a personalized one, but couldn't local site/physician features and interaction terms be added to a basic logistic regression model to achieve similar benefits?

Decent survey of related studies understanding clinical decision making factors, though with varying purposes.
In addition to review of different technical approaches, useful review of the live decision making process. I was aware of the MELD prioritization scheme for liver transplant offers, but not aware of the variability in organ offer acceptance rates that the authors focus their modeling on here.

Well laid out description of decision policy modeling framework, and reference to how that fits in the real-world process.


**Time Spent Reviewing:**

1

---

> ### Author Response · Authors · 2021-08-09
> **Response to Reviewer PPKT**
>
> Thank you for your insightful comments and questions. We give answers to each in turn.
>
> [ 1. Usefulness to decision makers ]
>
> (1.1) "The goal is not achieving a particular measurable outcome (e.g., prediction accuracy)":
> We agree with the reviewer that it is difficult to quantify the value or benefit that interpretability of suggestions can offer to decision makers. However, we would like to emphasize that to achieve our goal of discovering potential drivers of human decisions, high prediction accuracy is necessary. The personalized decision policies and improved prediction performance, while preserving similar interpretability to white-box models like logistic regression, are two of the major benefits of our method compared to logistic regression models.
>
> In addition, we have conducted a short survey with six clinicians in an attempt to further validate the usefulness of our method. The survey was conducted blindly, i.e., the clinicians were given no context as pertains to our proposed method. We asked clinicians several questions regarding the interpretability and usefulness of several different types of interpretation. Notably, all six clinicians deemed individualized logistic regression models more informative and practical than a general logistic regression model. In addition, five out of the six surveyed clinicians consider logistic regression models more interpretable and helpful than post-hoc interpretations of black-box models.
>
> Despite the relatively small sample size, we believe that these results provide evidence of the importance of interpretability in our proposed approach and the usefulness of our method compared to alternate forms of interpretability, including basic logistic regression models.
>
> [ 2. Local site/physician features and interaction terms ]
>
> We agree that including local site/physician features and extra interaction terms could improve the performance of basic logistic regression models. To test the reviewer’s hypothesis, we have evaluated a new baseline where we have augmented a basic logistic regression model with interaction terms. All polynomial combinations of feature maps in the original criteria space with degree two (i.e., pairwise interaction terms) are included as additional criteria for decision prediction. Part of the updated benchmark results with the new baseline included are given in the table below.
>
> | Method                                                              | AUC-ROC    | AUC-PRC     | NLL               |
> |:--------------------------------------------------------------|:-----------------|:------------------|:------------------|
> | Basic Logistic Regression                                 | 0.794±0.054 | 0.341±0.061 | -0.538±0.051 |
> | Basic Logistic Regression + Interaction Terms | 0.811±0.057 | 0.374±0.077 | -0.512±0.060 |
> | BC                                                                     | 0.904±0.040 | 0.523±0.062 | -0.370±0.067 |
> | iTransplant                                                        | 0.896±0.044 | 0.501±0.060 | -0.394±0.072 |
>
> Although the prediction performance of a basic logistic regression model is improved slightly with additional interaction terms, a significant gap in performance to both behavioral cloning (BC) and iTransplant remains. In comparison, our proposed method, iTransplant, achieves similar performance to the black-box BC model *without* any interaction terms.
>
> Please note that following the suggestion of Reviewer oN9u, we performed a more exhaustive hyperparameter selection for each model considered in our paper, which accounts for any minor deviations in these results compared to our original submission. The new hyperparameter selection is conducted in the same way as previously. Details can be found in (2.4) of our response to Reviewer oN9u.
>
> [ 3. Limitations and societal impact]
>
> Thank you for the insightful comments on the limitations and societal impact of our method. We will ensure this is discussed in greater detail in our final manuscript.
>
> (3.1) "Misinterpreting correlations found through these approaches as causal could negatively impact transplant decisions without clear guidance on interpretation of findings":
> As noted by the reviewer, the insights provided by our method could be used to positively impact organ transplant practices. However, we agree that correlations between features in the criteria space and decisions identified by our method should not be interpreted as causal, and care must be taken when interpreting the output of any inverse decision-making approach, including our method. We agree that it would be beneficial to expand our discussion of these points in our final manuscript.

---

### Decision · Program_Chairs · 2021-09-27

**Decision:**

Accept (Poster)

**Comment:**

Reviewers' scores and sentiments were remarkably consistent and positive toward this work.  The approach taken to address a high-stakes problem seems appropriate, and reviewers appreciated both that approach and its validation.  Still, reviewers -- especially Q9SD and oN9u -- did have outstanding questions regarding details about the methodology and about the experimental validation, and I would encourage the authors to update their work to reflect answers to those questions (many of which were given in the rebuttal).